# Climate sensitivity and meridional overturning circulation in the late Eocene using GFDL CM2.1

David K. Hutchinson[1], Agatha M. de Boer[1], Helen K. Coxall[1], Rodrigo Caballero[2], Johan Nilsson[2] and Michiel Baatsen[3]

[1]Department of Geological Sciences, Stockholm University, 10691 Stockholm, Sweden
[2]Department of Meteorology, Stockholm University, 10691 Stockholm, Sweden
[3]IMAU, Utrecht University, Princetonplein 5, 3584CC Utrecht, the Netherlands

*Correspondence to*: David K. Hutchinson ([david.hutchinson@geo.su.se](mailto:david.hutchinson@geo.su.se))

**Abstract.** The Eocene-Oligocene Transition (EOT), approximately 34 Ma ago, is an interval of great interest in Earth's climate history, due to the inception of the Antarctic ice sheet and major global cooling at the time. Climate simulations of the transition are needed to help us interpret proxy data, test mechanistic hypotheses for the transition, and determine the climate sensitivity at the time. However, model studies of the EOT thus far typically employ control states designed for a different time period, or ocean resolution on the order of 3°. Here we developed a new higher resolution paleoclimate model configuration based on the GFDL CM2.1 climate model adapted to a late Eocene (38 Ma) paleogeography reconstruction. The ocean and atmosphere horizontal resolutions are 1° x 1.5° and 3° x 3.75° respectively. This represents a significant step forward in resolving the ocean geography, gateways and circulation in a coupled climate model of this period. We run the model under 3 different levels of atmospheric $CO_2$; 400, 800 and 1600 ppm. The model exhibits relatively high sensitivity to $CO_2$ compared with other recent model studies, and thus can capture the expected Eocene high latitude warmth within observed estimates of atmospheric $CO_2$. However, the model does not capture the low meridional temperature gradient seen in proxies. Equatorial sea surface temperatures are too high in the model (30-37°C) compared with observations (max 32°C), though observations are lacking in the warmest regions of the western Pacific. The model exhibits bipolar sinking in the North Pacific and Southern Ocean, which persists under all levels of $CO_2$. North Atlantic surface salinities are too fresh to permit sinking (25-30 psu), due to surface transport from the very fresh Arctic (~20 psu), whose surface

salinities approximately agree with Eocene proxy estimates. North Atlantic salinity increases by 1-2 psu when $CO_2$ is halved, and similarly freshens when $CO_2$ is doubled, due to changes in the hydrological cycle.

## 1 Introduction

The Eocene-Oligocene Transition (EOT), approximately 34 Ma ago, was one of the major climate transitions in the Cenozoic Era. The EOT marks the shift from the so-called 'greenhouse world' to the 'icehouse world', during which time the first semi-permanent ice sheets formed on Antarctica, and major changes occurred in flora and fauna across the globe reflecting a shift to colder or drier conditions (Coxall and Pearson, 2007; Dupont-Nivet et al., 2007; Miller et al., 1991). Yet the causes of the transition and its global impacts are still not fully understood, particularly regarding the role of ocean circulation. The prevailing proposed mechanisms for the transition are: a decline in atmospheric $CO_2$ below a critical threshold (DeConto and Pollard, 2003), and the opening of Southern Ocean gateways causing thermal isolation of Antarctica (Kennett, 1977); with orbital forcing playing a key role as trigger and pacemaker of ice growth on Antarctica (Coxall et al., 2005). In all cases, ice-albedo feedbacks play an major role in enabling the beginning of land-ice formation to grow into a continental ice sheet. Paleogeography has also been found to partly control the sensitivity of these mechanisms, as illustrated by Lear and Lunt (2016).

A long-term decline in atmospheric $CO_2$ has been found to be a major driver of Eocene cooling (Anagnostou et al., 2016), and a plausible trigger of the EOT glaciation (DeConto and Pollard, 2003). Hereafter we refer to "atmospheric $CO_2$" as simply $CO_2$. This mechanism is appealing in its simplicity, however the threshold required to trigger the glaciation is uncertain. Gasson et al. (2014) found the glaciation threshold to be within a range of 560 to 920 ppm using an intercomparison of late Eocene climate models to force an ice sheet model. This large range was primarily due to the spread in climate sensitivity and seasonality of the models, of between 2.5°C and 4.1°C per doubling of $CO_2$ under modern conditions. Observations of $CO_2$ from just prior to the EOT are similarly uncertain, with estimates from carbonate microfossils providing an uncertainty range of 450 to 1500 ppm with a central

estimate of 760 ppm (Pearson et al., 2009). Strong $CO_2$ forcing has had considerable success in explaining the very warm climate of the early Eocene (Huber and Caballero, 2011). The subsequent increase in seasonality and meridional temperature gradient from the early to late Eocene supports the idea that $CO_2$ forcing had a primary role in Eocene to Oligocene climate change (Eldrett et al., 2009).

The evolution of the oceanic meridional overturning circulation (MOC) around the EOT remains a topic of debate. Global compilations of benthic foraminiferal stable isotopes of $\delta^{13}C$ and $\delta^{18}O$ have been interpreted to indicate that the deep ocean was largely homogeneous during the early to middle Eocene (Cramer et al., 2009). This has been used to infer a predominance of southern hemisphere high-latitude

deep-water formation. Subsequent divergence of inter-basin benthic isotopic gradients creating oceanic heterogeneity has been linked to onset of North Atlantic intermediate water formation in the late Eocene, and the beginning of a more mature North Atlantic deep water formation in the early Oligocene (Borrelli et al., 2014). In those studies, the proposed forcing mechanism of North Atlantic deep water is the opening of Southern Ocean gateways. This notion rests on the modern-climate interpretation that, in

addition to diapycnal mixing in the interior ocean (Munk and Wunsch, 1998), wind-driven upwelling in the ACC is a major driver of the deepwater cell presently associated with the Atlantic MOC (Toggweiler and Samuels, 1995). However, transports by mesoscale eddies counter the wind-driven upwelling in the ACC and the strength of the MOC (Marshall and Speer, 2012). Recent modelling suggests that deepening of the Greenland-Scotland Ridge may be a trigger for the onset of North

Atlantic sinking (Abelson and Erez, 2017), and the resulting changes in heat transport could also have important effects on the period's climate.

In contrast to the North Atlantic sinking hypothesis, Nd isotope distributions suggest that North Pacific sinking occurred all the way from the Cretaceous through to the Miocene (Ferreira et al., 2018). Nd

tracer-enabled ocean simulations that are able to reproduce the observed pattern of Eocene Nd have vigorous bipolar Pacific sinking, with southern rather than North Pacific bottom water dominating (Thomas et al., 2014). Furthermore, idealized GCM simulations show that salt advection feedbacks create a competition between North Pacific and North Atlantic deep water formation (Wolfe and Cessi,

2014), so that the onset of North Atlantic sinking may imply a shutdown of North Pacific sinking and vice versa. It remains unclear if such a switch occurred at the EOT, and its potential triggers have not been fully explored.

Modelling studies investigating the effect of the Antarctic Circumpolar Current (ACC) on global climate have found that opening the Drake Passage leads to cooling of the Southern Ocean and a reduction in poleward heat transport towards Antarctica (Cox, 1989; Sijp et al., 2009; Sijp and England, 2004). These studies used either ocean-only or intermediate complexity climate models lacking dynamic atmospheric feedbacks. The Antarctic response of fully-coupled climate models to Drake

Passage opening is less clear, since the opening of a circumpolar gateway does not guarantee a strong Antarctic Circumpolar Current under late Eocene paleogeography (Zhang et al., 2010), or under the high $CO_2$ (Lefebvre et al., 2012) prevalent in the late Eocene. Furthermore, there is uncertainty of up to tens of millions of years over when a deep gateway first existed in Drake Passage (Barker et al., 2007; Livermore et al., 2005), so that the inception of the ACC cannot be easily pinpointed to the EOT. The

opening of a deep Tasman Seaway has also been suggested to occur around the EOT and cause Antarctic glaciation (Stickley et al., 2004). A model study of this opening found that the gateway change had a limited effect on Antarctic surface climate (Sijp et al., 2011). However, the gateway change may explain ~3°C of deep ocean cooling (Sijp et al., 2014), in agreement with late Eocene cooling found in deep sea foraminifera (Bohaty et al., 2012). Part of the $CO_2$ decline at the EOT may

have been caused by the opening of Southern Ocean gateways (Elsworth et al., 2017), due to feedbacks in silicate weathering. This presents a possible way to reconcile the $CO_2$ forcing and gateway hypotheses.

A common shortcoming of previous EOT simulations is an unrealistic representation of topography (by

'topography' we mean both land topography and ocean bathymetry). Many studies investigating gateway effects have used modern topography for their control simulation (Elsworth et al., 2017; England et al., 2017; Fyke et al., 2015; Sijp et al., 2009; Yang et al., 2014) and created EOT-like perturbations by closing Drake Passage and/or opening the Panama Seaway. While this method of

changing basin geometry is appealing in that it illuminates specific gateway effects, it implicitly ignores several features of the late Eocene or early Oligocene paleogeography that are potentially crucial to the ocean circulation. These include particularly the closed Pacific-Arctic Bering Sea seaway, the narrower Atlantic Basin, the shallow connections between the Atlantic and the Arctic, and the connections of the

Tethys Ocean with the Indian and Arctic Oceans (Figure 1). Other modelling studies of the EOT have been configured with early Eocene (~55 Ma) paleogeography. These have particularly focused on the effects of changing $CO_2$ forcing from the Eocene to the Oligocene (Eldrett et al., 2009; Liu et al., 2009), the impact of an ice sheet on the atmosphere and ocean circulation (Goldner et al., 2013, 2014), and the impact of opening Southern Ocean gateways (Goldner et al., 2014). While early Eocene topography is a

much closer starting point to late Eocene topography than modern, early and late Eocene topographies differ significantly due to ~20 Ma of tectonic evolution. This includes the gradual opening of the Tasman Gateway and Drake Passage, and the widening of the Atlantic.

Some recent climate model studies have used reconstructions of late Eocene and early Oligocene

topography for their control runs (Kennedy et al., 2015; Ladant et al., 2014; Lefebvre et al., 2012; Lunt et al., 2016; Zhang et al., 2010). Kennedy et al. (2015) found that when an ice sheet is placed over Antarctica, the Pacific sector of the Southern Ocean surface warms significantly when the Tasman Seaway is constricted as in the late Eocene, whereas when the Tasman Seaway is wider as in the early Oligocene, there is instead a weak cooling. These results contrast with Goldner et al. (2014), who found

a strong cooling of the Southern Ocean in response to Antarctic glaciation, albeit with an early Eocene paleogeography. Thus, there remains uncertainty over whether the opening of Southern Ocean gateways caused major cooling, or whether the glaciation itself was the primary driver of Southern Ocean temperature change. A summary table of previous EOT climate simulations is shown in Table 1.

| Model | Ocean Resolution | Atmos. Resolution | Age of Paleogeography | Sinking Regions | Reference |
|---|---|---|---|---|---|
| CCSM1.4 | 1.8° x 3.6° | 3.75° | Early Eocene (Sewall et al., 2000) | Southern Ocean and N. Atlantic | (Huber et al., 2004; Huber and Sloan, 2001) |
| CCSM3 | 1° x 2.8° | 3.75° | Early Eocene (Sewall et al, 2000) | Southern Ocean | (Eldrett et al., 2009; Liu et al., 2009) |
| CESM1.0 | 1.8° x 3.6° | 3.75° | Early Eocene (Sewall et al, 2000) | Southern Ocean | (Goldner et al., 2014) |
| CESM1.0.5 | 1° | 2° | Late Eocene 38 Ma (Baatsen et al, 2016) | S. Pacific only | (Baatsen et al., 2018a) *in review* |
| CM2Mc | 2.5° x 0.6-3° | 3° x 3.75° | Modern + gateway perturbations | Southern Ocean and N. Atlantic | (Elsworth et al., 2017; Yang et al., 2014) |
| COSMOS | 1.8° x 3° | 3.75° | Early Miocene 20-15 Ma (Herold et al., 2008) + gateway perturbations | Southern Ocean and N. Atlantic | (Stärz et al., 2017) |
| FOAM | 1.4° x 2.8° | 4.5° x 7.5° | Early Eocene (Zhang et al., 2010) | Not stated | (Zhang et al., 2010) |
| FOAM | 1.4° x 2.8° | 4.5° x 7.5° | Middle Oligocene (Scotese, 2001) | Southern Ocean | (Ladant et al., 2014; Lefebvre et al., 2012) |
| HadCM3L | 3.75° x 2.5° | 3.75° x 2.5° | - 38 to 34 Ma Rupelian - 34 to 30 Ma Priabonian (GETECH – proprietary) | Southern Ocean and N. Atlantic | (Inglis et al., 2015; Kennedy et al., 2015) |
| MITgcm (ocean only) | 4° | N/A | Early Eocene (Sewall et al, 2000) | S. Pacific and N. Pacific | (Thomas et al., 2014) |
| POP2.1 (ocean only) | 1° | N/A | Late Eocene 38 Ma (Baatsen et al, 2016) | S. Pacific or N. Pacific | (Baatsen et al., 2018b) |
| UVic | 1.8° x 3.6° | EBM | Modern + gateway perturbations | Southern Ocean and N. Atlantic | (Fyke et al., 2015) |
| UVic | 1.8° x 2.4° | EBM | Early Eocene (Huber et al, 2004) | Southern Ocean | (Sijp et al., 2011) |
| **GFDL CM2.1** | **1° x 1.5°** | **3° x 3.75°** | **Late Eocene 38 Ma (Baatsen et al, 2016)** | **Southern Ocean and N. Pacific** | **This study** |

**Table 1: A non-exhaustive summary of previous EOT climate simulations, including their ocean and atmosphere horizontal resolutions, age of paleogeography and sinking regions. EBM represents a 2D energy-balance model. All are coupled climate models except the MITgcm (Thomas et al., 2014) and POP2.1 (Baatsen et al., 2018b) simulations,**
5 **which are included because they specifically investigate the meridional overturning circulation (MOC) around the EOT.**

Model studies of the EOT, including those mentioned above, generally employ very low - typically 3° - horizontal ocean resolution. This is often dictated by computational constraints, since the deep ocean
10 takes thousands of years to equilibrate (Danabasoglu et al., 1996). Furthermore, uncertainties in the reconstructed paleogeography often place limitations on the fidelity of the relevant topographic features. We are interested in improving the ocean resolution for two reasons. First, the Arakawa B-grid used

ubiquitously in paleoclimate model studies requires narrow ocean straits to be at least 2 grid cells wide, in order to have a non-zero velocity grid point. If a model uses 3° resolution, then the minimum width of a strait is 6°, corresponding to 670 km in the meridional direction. In the zonal direction, 6° of distance varies from 670 km at the equator to 330 km at 60° latitude. These distances are large enough

that key EOT ocean gateways, such as Drake Passage and Arctic gateways will hardly be resolved. Second, there is evidence that gateway effects are enhanced at higher resolution (Viebahn et al., 2016), especially in controlling the heat transport of boundary currents. For example, moving from 1° to 0.25° (eddy-permitting) resolution in the modern-day climate results in an enhanced western boundary current heat transport (Delworth et al., 2012), and this effect can cause substantially lower Arctic sea ice

coverage (Hutchinson et al., 2015; Kirtman et al., 2012). Moving from 3° to 1° ocean resolution, while still not permitting eddies, would improve the representation of boundary currents and their associated impacts on gateway transitions.

This study presents a new fully-coupled climate simulation of the late Eocene using order 1° ocean

resolution and a state-of-the-art paleotopographic reconstruction. We apply the late Eocene (38 Ma) reconstruction of Baatsen et al. (2016) to generate topography for the GFDL CM2.1 model, as shown in Figure 1. This ensures that global tectonic evolution relevant to the late Eocene is included. Within this framework, we explore the climate sensitivity to $CO_2$ perturbations, namely using 400, 800 and 1600 ppm (approximately 1.4x, 2.9x and 5.7x preindustrial $CO_2$ respectively). We also explore the sensitivity

to changing the parameterized ocean vertical mixing from the standard Bryan-Lewis scheme to a tidal-mixing scheme where the diffusivity is set to a constant background value and enhanced near the bottom. We examine the impact of these changes on the oceanic MOC and global climate.

## 2 Model Description

### 2.1 Model Components

This study uses a modified version of the coupled climate model GFDL CM2.1 (Delworth et al., 2006), using boundary conditions for the late Eocene. The ocean component is updated to the modular ocean

model (MOM) version 5.1.0, while the other components of the model are the same as in CM2.1, namely Atmosphere Model 2, Land Model 2 and the Sea Ice Simulator. In the ocean and sea ice components, the horizontal grid is modified to have a resolution of 1° latitude x 1.5° longitude. We use a tripolar grid as shown in Figure 7 of Murray (1996), with a regular latitude-longitude mesh south of

65°N, and a transition to a bipolar Arctic grid north of 65°N. The poles then lie over North America and Siberia to avoid convergence of meridians in the Arctic Ocean. This enables the model to simulate Arctic Ocean velocities without damping in the vicinity of the North Pole. Unlike in CM2.1, there is no refinement of the latitudinal grid spacing in the tropics. The ocean retains the original 50 vertical levels with the same grid spacing as in CM2.1. The atmospheric horizontal grid resolution is 3° x 3.75°, with

24 vertical levels. This atmosphere grid is identical to that used in CM2Mc (Galbraith et al., 2010). This choice of grid resolutions gives good load balancing between the atmosphere and ocean components, while enabling better resolution of coastlines and straits than most existing Eocene models (~3° ocean resolution).

The topography (both land and ocean) uses the late Eocene (38 Ma) reconstruction of Baatsen et al., (2016). This topography is distinct from previous reconstructions (e.g. Markwick, 2007) in that it uses a Paleomagnetic reference frame to position the continents (van Hinsbergen et al., 2015; Torsvik et al., 2012), rather than a Hotspot reference frame (Seton et al., 2012). Continental elevation is based on ETOPO modern day topography, which is then relocated to its 38 Ma position by plate tectonic motion

using GPlates. For the deep ocean, an age-depth relationship is applied (Müller et al., 2008) and adjusted to the Paleomagnetic reference frame. Manual adjustments are then applied to areas where elevation changes are well constrained by geological evidence. Specific regions of adjustment include Antarctica, the Himalayas, the Amazon, Turgai Strait and the Tethys Sea. Where paleo-elevation data is missing or unknown, the topography defaults to modern-day elevation. One region of uncertainty is in

the gateway between the Arctic and Atlantic Ocean, where other reconstructions have a more constricted throughflow (e.g. Figure 41 of Markwick, 2007). In particular, deepening of the Greenland-Scotland Ridge is hypothesised as a trigger of North Atlantic deep water formation at the EOT, via its

impact on freshwater transport from the Arctic into the Atlantic (Abelson and Erez, 2017; Stärz et al., 2017).

Manual adjustments were made to the topography to ensure that all straits are at least 2 grid cells wide, so that they have corresponding velocity grid points, and no ocean grid cell is isolated. The resulting ocean bathymetry is shown in Figure 1a. In the atmosphere, the topography is interpolated onto the horizontal grid and then smoothed using a 3-point mean filter to ensure a smoother interaction with the wind field. This filtering was mainly needed on the Antarctic continent, due to convergence of meridians on the topography grid, which caused numerical noise in the wind field during initial testing. This filter will likely be removed outside of the polar regions in future versions of the model. The resulting topography is shown in Figure 1b. Vegetation types are based on a dataset based on Sewall et al. (2000), with modifications where data are available from Thorn and DeConto (2006); Utescher and Mosbrugger (2007); and Gomes Rodrigues et al. (2012). The Sewall et al. (2000) dataset was originally configured for the CESM plant functional types, and we have adapted these to the corresponding vegetation type in CM2.1. The river runoff is determined by a relocation map, where each land point is assigned a corresponding coastal location for returning runoff to the ocean. This relocation map was determined by a downslope relocation algorithm from the model topography. Aerosol forcing was taken from the Eocene (55 Ma) reconstruction of Herold et al. (2014), and adapted to our model's input types.

## 2.2 Vertical Mixing Scheme

We use a simplified version of the tidal mixing scheme of Simmons et al. (2004). The CM2.1 code distribution provides a heterogeneous distribution of seafloor roughness amplitude based on high resolution maps of the modern seafloor. We set this to be uniform for simplicity, since estimating the late Eocene seafloor roughness is not straightforward. We thus calculate an average roughness amplitude of 210.512 m (the $h$ parameter from Eq. (1) of Simmons et al. 2004), and set a background diffusivity of $1.0 \times 10^{-5}$ $m^2$ $s^{-1}$. In parallel we also simulate the model using the standard Bryan-Lewis diffusivity in CM2.1, commonly used in deep time paleoclimate studies (e.g. Lunt et al. 2012). We

prefer a bottom roughness mixing scheme to Bryan-Lewis, since the former is more physically realistic. The Bryan-Lewis scheme uses the following values:

$$K_{eq}(z) = 10^{-4} \left[ 0.65 + \frac{1.15}{\pi} \tan^{-1}(4.5 \times 10^{-3}(z - 2500)) \right]$$

$$K_{pole}(z) = 10^{-4} \left[ 0.75 + \frac{0.95}{\pi} \tan^{-1}(4.5 \times 10^{-3}(z - 2500)) \right]$$

5    Where $K_{eq}$ and $K_{pole}$ are the equatorial and polar diffusivities respectively, in $m^2$/s. The depth $z$ is in metres, which has a transition level at 2500 m as shown above. The low latitudes use $K_{eq}$ and the high latitudes use $K_{pole}$, with a transition at 35° latitude.

## 2.3 Initial Conditions and Spinup

The initial conditions of the ocean model are configured to a modified version of the DeepMIP
10    experimental design (Lunt et al., 2017). The temperature and salinity are set to the following profiles:

T [°C] = (5000 – z) / 5000 * 20 cos ($\phi$) + 10 ;   if $z \leq$ 5000 m.

    = 10 ;   if $z >$ 5000 m.

S [psu] = 35.0

where $\phi$ is latitude, z is depth of the ocean (metres below surface – positive downwards).

The model is run at three different levels of $CO_2$: 400, 800 and 1600 ppm, where 800 ppm is deemed as the control simulation. The atmosphere and land surface components are initialised from a previous test simulation.

We used the following procedure to spin up the model simulations:

1.  The model is run in coupled mode for 50 years.
2.  The last 10 years of this coupled simulation are used to generate surface boundary conditions for an ocean and sea ice only simulation.

3. The model is run in ocean and sea ice only model for 500 years, using a 10-year repeating pattern of forcing based on the CORE protocol (Griffies et al., 2009). During this simulation, we doubled the ocean tracer time step with respect to the momentum time step.

4. The resulting ocean state is used to re-initialise step 1.

5. After 6 cycles of the above (i.e. a total of 3300 model years), the simulation was continued in fully-coupled mode (with no acceleration of tracers) for a further 3200 years.

In order to assess the robustness of this acceleration method, we also ran the control simulation from the same initial conditions for 1000 years in coupled mode only. This coupled-only simulation yielded an ocean state that was very similar in key metrics (i.e. temperature, salinity, age tracer) to the iteratively coupled ocean run at year 2000 (see Figure 2). In other words, the iterative coupling procedure achieved a climate state that was similar to that of an ordinary coupled run, with roughly twice the number of model years needed to reach the same state. This approximately cancels out the computational speed-up achieved by the decoupling procedure. We therefore abandoned the iterative coupling procedure after 6 cycles and completed the spinup in coupled mode.

The spinup evolution of ocean temperature across the three levels of $CO_2$ is shown in Figure 2. The simulations all have surface climates in quasi-equilibrium, though the deep ocean is gradually cooling in all cases. All simulations have a temperature trend of less than 0.1°C per century at 4000 m, though the warmer climates are trending more slowly than the colder ones. This is because the initial conditions are designed to be warmer rather than colder than equilibrium, so that convection can readily occur due to surface cooling. The step changes in SST over the first 3300 years are due to the coupling/decoupling procedure described above. We also examined the evolution of the meridional overturning circulation during spinup (Figure 2e). This shows that the magnitude of Northern Hemisphere overturning remains steady over the last ~2000 years, while the Southern Hemisphere overturning is gradually reducing in the 1600 ppm and 400 ppm cases, and is steady in the 800 ppm case. Southern Ocean sinking is discussed further in Section 3 and 4. The trend in Arctic salinity is also shown in Figure 2f. This

indicates a rapid adjustment towards a fresh Arctic surface in all three cases, followed by minimal change over the last 3000 years.

One way to estimate the final equilibrium in a transient climate simulation is to derive a 'Gregory plot'

(Gregory et al., 2004), which compares the top-of-atmosphere (TOA) net radiative forcing with the temperature anomaly. This is shown for each $CO_2$ level in Figure 3. The data points are derived from 10-year averages during the iterative coupling phase of the spinup, and then from 100-year averages during the fully coupled phase. There is a trend towards zero net radiative forcing during much of the spinup, but the curves begin to flatten before they reach zero. This makes extrapolation towards an

equilibrium value difficult, since there is no definitive slope from which to project. In addition, we note the CM2.1 model has been documented to have a radiative imbalance under pre-industrial conditions on the order of 0.5 $W/m^2$ (Delworth et al., 2006). This is due to a number of factors, including a systematic loss of energy (~0.3 $W/m^2$) in the atmosphere model, and a heat sink that occurs when water is returned to the ocean from precipitation or runoff (~0.14 $W/m^2$). We therefore do not expect the model to reach a

perfect radiative forcing balance at equilibrium.

## 3 Climate and Ocean Circulation

The late Eocene paleogeography incorporates changes that affect important features of the ocean circulation compared with the present day. The Southern Ocean gateways of Drake Passage and the Tasman Seaway are narrow and shallow (sill depths of around 500 m), and permit a weak eastward

circumpolar flow. Panama Seaway is open, and allows a shallow connection of surface water between the equatorial Atlantic and Pacific Oceans. The Bering Strait is closed, so that the North Pacific has no direct connection with the Arctic Ocean. The Arctic Ocean is connected to the North Atlantic by narrow and shallow seaways (up to 200 m depth), which allow export of freshwater from the Arctic into the Atlantic. Transport between the Arctic and Atlantic is bidirectional, with surface waters travelling

predominantly southwards, and waters below 100 m travelling predominantly northwards. The Tethys Ocean has connections to the Indian and the Atlantic, and a very narrow and shallow connection via Turgai Strait to the Arctic (25 m depth), allowing unidirectional southward transport of surface waters.

Apart from these gateway changes, the Pacific is wider and more open through the Indonesian Archipelago, while Australia is further South. The Atlantic is narrower especially in the Northern Hemisphere, and in this region the subpolar gyre is particularly constricted. These features all contribute to a vastly different ocean circulation than present day, and we explore the major changes below.

## 3.1 Surface Climate

In this section, we describe the control climate (800 ppm), key features of the circulation and how they compare with proxy evidence. Figure 4a and Figure 4b show the sea surface temperature (SST) and surface air temperature (SAT) respectively. Equatorial SSTs reach as high as 38°C in the western Pacific warm pool, with tropical temperatures around 30-35°C elsewhere. The warm pool extends west into the Indian Ocean due to an open Indonesian Seaway. The meridional temperature gradient is a little lower than in the modern climate, with the high latitude SSTs of the North Pacific and Southern Ocean being around 15-20°C, even along the coast of Antarctica.

The sea surface salinity (Figure 4c) indicates very fresh conditions in the Arctic Ocean, around 20 psu. The fresh Arctic surface waters flow out primarily into the North Atlantic, where the surface salinities around Greenland and above ~45°N are generally below 30 psu. Under these conditions deep sinking does not occur in the North Atlantic. On the other hand, the high latitude North Pacific surface water is mostly around 35 psu, apart from a limited region in the north-east where there is strong runoff entering the ocean from the North American mountain ranges. Under modern conditions, orography preferentially freshens the surface water in the North Pacific over the North Atlantic (Maffre et al., 2017; Sinha et al., 2012; Wills and Schneider, 2015). This is primarily due to its influence on precipitation and drainage pathways, especially in the North American mountain ranges. Notably, the low surface salinities prevent deep water formation in the present-day North Pacific (Warren, 1983). However, our control simulation shows a higher river runoff (0.28 Sv) going into the Arctic than the North Atlantic (0.14 Sv) and North Pacific (0.15 Sv) (here the cutoff latitude is chosen to be 45°N). The same comparison with a simulation of CM2.1 in its modern configuration produces a river runoff of 0.10 Sv into the Arctic, only slightly higher than North Atlantic of 0.07 Sv and North Pacific of 0.08 Sv.

This high Arctic runoff, combined with enhanced P-E forcing of the warm climate creates the very fresh Arctic surface waters. Transport of this fresh Arctic water mass into the North Atlantic causes a state reminiscent of the present day North Pacific with low surface salinity, a strong halocline and no deep water formation.

Zonal wind stress (Figure 4e) in the Southern Ocean has a more zonal structure than in the present day, with a weaker standing wave meander and a lower magnitude overall. The wind stress peak sits around 50°S, similar to present day, and the wind stress curl drives subtropical gyre circulations in the South Pacific and Indian Oceans analogous to the present day, as shown in the barotropic streamfunction of

Figure 4f. The Southern Ocean gateways — Drake Passage and the Tasman Seaway — are open, but they are narrow and shallow and thus allow only a weak circumpolar flow around Antarctica of 15 Sv.

The mean state of the model has an east-west temperature contrast across the equatorial Pacific in excess of 6°C, slightly larger than in the present day. Empirical orthogonal function (EOF) analysis of

global SST indicates strongest variability in the central equatorial Pacific (Figure 5a), while precipitation varies most strongly in a dipole spanning the western equatorial Pacific and Indian oceans (Figure 5b). Using the first EOF of SST, we define an 'El Niño Index' region between 170°E and 140°W, and 5°S to 5°N, as being representative of highest Pacific equatorial SST variability. This index is analogous to the present-day Niño-3.4 index, designed to bracket the region of highest El Niño-

Southern Oscillation (ENSO) variability. This index indicates the occurrence of multi-year El Niño and La Niña events (Figure 5c), defined by a threshold of +0.5°C for an El Niño and -0.5°C for a La Niña, as used in present day. The magnitude of temperature deviations is generally weaker than present day Niño-3.4 variability. The east-west SST gradient in the equatorial Pacific is closely anti-correlated with the El Niño index, with a reduction in east-west gradient during El Niño and vice versa for La Niña

(Figure 5d). However, the magnitude of these variations is also weaker than present day. The areal extent of the western Pacific warm pool in our model is larger than the present day. We suggest that this partly due to fewer land barriers in the western Pacific and a wider basin. We argue that this creates a larger thermal inertia in the western Pacific, implying a reduction in variance in the east to west thermal

gradient. This result agrees broadly with von der Heydt and Dijkstra (2006), who also found a stronger western Pacific warm pool in an Oligocene model simulation.

The 'Weddell Sea' equivalent in the EOT model, where deep water forms in the model, has annual mean sea surface temperature below 10°C and develops a small area of seasonal sea ice close to the coast. The Arctic Ocean temperatures are typically around 4-6°C, and it is mostly ice free all year round, apart from some embayments around the Siberian coast. The average minimum monthly sea ice thickness for each hemisphere is plotted in Figure 6, showing September thickness for the Northern Hemisphere and March thickness for the Southern Hemisphere. We note that there is evidence of ice-rafted debris in the Arctic from the middle Eocene onwards (Stickley et al., 2009), indicating the likely presence of seasonal sea ice at this time. Our 400 ppm simulation has Arctic-wide sea ice in summer (Figure 6b), in agreement with this proxy evidence. However, the ice-rafted debris emanates from the central Arctic (Stickley et al., 2009), where there is still some sea ice in the 800 ppm case, making it difficult to distinguish which case is more realistic.

## 3.2 Meridional Overturning Circulation

The meridional overturning circulation (Figure 7) shows a structure of sinking in the North Pacific and Southern Ocean. There is no deep overturning cell in the North Atlantic, due to the surface waters being far too fresh to allow for local deep water formation. The structure of the Pacific overturning is analogous to modern day North Atlantic Deep Water formation: the Northern Component Water is warmer and saltier than the Southern Component water and is heavier at the surface. This difference in deep water properties is caused by colder winter temperatures in the Southern Ocean sinking regions than the North Pacific, since deep water forms exclusively in winter. The colder winter temperatures are due to the asymmetry of landmasses between the poles. One factor is that the Southern Ocean extends further poleward than the North Pacific (in both Pacific and Atlantic sectors). Furthermore, the Antarctic continental interior becomes colder in winter than the Arctic Ocean, and therefore the seasonal cycle in high latitude (>60°) surface air temperature is stronger in the Southern Hemisphere.

The Northern Hemisphere also has very cold winters in continental interiors, but its polar winters are moderated by the presence of an ocean and its seasonal cycle is milder.

As in modern climate, thermobaric effects control the layering of the abyssal ocean (Nycander et al., 2015). The Southern Hemisphere sourced deep waters are colder and therefore more compressible than the warmer and slightly more saline ones sourced from the Northern Hemisphere. Thus bottom waters are predominantly formed in the Southern Ocean. Sinking occurs in both the Pacific and Atlantic sectors of the Southern Ocean, as indicated by the age tracer at 2000 m and winter mixed layer depths (Figure 8). The shallow gateways of Drake Passage and Tasman Seaway inhibit the exchange of deep water between the basins, preventing Pacific- and Atlantic-sourced bottom waters from directly reaching the opposite ocean basin. The poleward heat transport is similar to modern (not shown), with the partition between ocean and atmosphere heat transport being dominated by the ocean in the tropics only, and by the atmosphere in the mid- to high- latitudes. As in the modern ocean, poleward heat transport is skewed towards the Northern Hemisphere, with a peak value of 2.2 PW (northward) and a minimum value of -0.9 PW in the Southern Hemisphere. We suggest that this asymmetry is due to the analogous asymmetry of deep water formation; i.e. the northern overturning cell traverses a larger temperature difference and hence enhances northward ocean heat transport relative to weaker temperature gradient across the southern overturning cell. In the tropics, we do not find any warm salt-driven deep water formation, in line with previous studies (Table 1).

We also compare the results of our control run with the widely-used Bryan-Lewis mixing scheme, as used in the default case of CM2.1 (Delworth et al., 2006). The Bryan-Lewis simulation was spun up using the same procedure as the control run, as described in Section 2.3. Prior to this study, we expected that changes to the bathymetry would alter the deep ocean mixing distribution due to intensified mixing over ridges and seamounts. Furthermore, we anticipated that changes to surface radiative forcing would alter the thermocline structure. For these reasons we expected that a bottom-enhanced vertical mixing scheme would be more adaptable to late Eocene climate conditions and perform better than the fixed vertical mixing structure of the Bryan-Lewis scheme. However, we find that the meridional overturning

circulation and stratification are largely insensitive to this change in vertical mixing. The magnitude of the overturning, distribution of sinking regions and age tracer were all very similar between the mixing schemes (not shown). This was a surprising result, given theoretical predictions of change due to bottom-enhanced mixing (de Boer and Hogg, 2014). We do find that the abyssal ocean is slightly cooler in the bottom-enhanced mixing scheme (0.5°C), suggesting a weaker diffusive heat penetration into the abyss, in line with theoretical predictions. We also find abyssal salinity differences on the order of 0.1 psu. Given the very long spinup time of the simulations, the lack of divergence between the schemes is surprising. In the Arctic Ocean, the differences in deep ocean temperature and salinity are greater. This is because the deep waters in the Arctic are very stably stratified by the halocline and can only communicate with the surface diffusively.

In order to examine the effective diffusivity of each mixing scheme, we computed the stratification-weighted diffusivity (de Lavergne et al., 2016) between the two schemes, i.e.

$$\frac{\iint K_v \, N^2 dx \, dy}{\iint N^2 dx \, dy}$$

Where $K_v$ is the total vertical diffusivity, and $N^2$ is the buoyancy frequency. This was computed offline using monthly output of $K_v$ and $N^2$ and averaged across the last 50 years of simulation, shown in Figure 9. This calculation shows that the effective mixing across the thermocline is indeed similar between the two schemes. Mixing is slightly stronger in the Bryan-Lewis scheme between 2500 and 3000 m, while mixing near the seafloor is indeed stronger in the bottom-enhanced scheme. There is also larger effective mixing in the Atlantic than the Indo-Pacific in both schemes, and also a larger difference between the bottom-enhanced mixing and the Bryan-Lewis scheme in the Atlantic.

We note further that the Simmons et al., (2004) mixing scheme used in our control run assumes that the energy input to the mixing within each grid column is proportional to the simulated bottom stratification. An increase in bottom stratification (e.g. as a result of increased surface density gradients) is thus immediately paralleled by an increase in mixing energy. This can have the effect of maintaining approximately constant diffusivities and circulation rates despite larger density contrasts. In this way,

the Simmons et al., (2004) scheme is similar to the Bryan-Lewis scheme in that it fixes diffusivity rather than mixing energy. A better approach may be to fix the energy consumed by vertical mixing, since energy constraints provide a physically consistent framework to relate the drivers of mixing to the abyssal circulation (de Lavergne et al., 2016).

## 3.3 Comparison with Proxy Data

A comparison of proxy data with the model SSTs for the 400, 800 and 1600 ppm experiments is shown in Figure 10. The data are taken from the SST proxy compilation from 38 to 34 Ma shown in Table A2 of Baatsen et al., (2018a). The compilation combines UK'$_{37}$ , TEX$_{86}$, Mg/Ca, $\delta^{18}$O and $\Delta_{47}$ data (Bijl et al., 2009; Douglas et al., 2014; Evans et al., 2018; Hines et al., 2017; Kamp et al., 1990; Kobashi et al., 2004; Liu et al., 2009; Okafor et al., 2009; Pearson et al., 2001, 2007; Petersen and Schrag, 2015; Tripati and Zachos, 2002; Wade et al., 2012) The paleolocations of these proxies are consistent with the model boundary conditions (Baatsen et al., 2016). Figure 10 shows the proxy data points compared with the nearest model ocean point, and the model zonal mean temperature. The 1600 and 800 ppm cases are too warm compared with the proxies in the tropics. In the control case (800 ppm) the model is roughly 3°C warmer at the equator. However, the TEX$_{86}$ data from Tanzania (~32°C) of Pearson et al. (2007) agree better with the control case. In the mid to high latitudes, the control case is colder than the proxy estimates. The 1600 ppm case agrees better with the mid to high latitude proxy estimates, however its warm bias at the equator is very large. In the 400 ppm simulation, the tropical temperatures agree well with the proxies, while in the mid-latitudes, it is too cold.

The model's inability to capture the low meridional temperature gradient in the proxies is a common problem in simulating warm climates of the Eocene (Huber and Caballero, 2011). However, several aspects of the proxy data could help to improve this situation. First, alternative calibrations of the TEX$_{86}$ data from Tanzania may yield temperatures as high as 35°C in the late Eocene (Bijl et al., 2009), which would reduce the low gradient problem. We also note that there are no available proxy estimates from 38 to 34 Ma in the western Pacific warm pool, where the ocean's warmest temperatures are found. Middle Eocene (42 to 38 Ma) SST estimates from Java are as high as 35-36°C (Evans et al., 2018).

Second, high latitude TEX$_{86}$ data may be affected by summer bias due to seasonal growth of its underlying species (Kim et al., 2010), which may have the effect of reducing the meridional temperature gradient seen in the proxies.

## 4 Climate Sensitivity to CO$_2$

### 4.1 Temperature Response

The global mean SST in the control run (800 ppm) is 27.8°C; halving CO$_2$ to 400 ppm produces 3.5°C of cooling, and doubling CO$_2$ to 1600 ppm yields 4.2°C of warming. The global mean SAT in the control is 25.6°C; halving CO$_2$ to 400 ppm produces 4.0°C of cooling, and doubling produces 4.8°C of warming. The higher sensitivity of temperature change in the atmosphere is due to both stronger polar

amplification in the atmosphere than the ocean; and a stronger temperature change over land than over the ocean. In its modern configuration, CM2.1 has an equilibrium sensitivity of 3.4°C (Winton et al., 2010), though we note that this sensitivity may be different when using the present lower resolution atmospheric-model component. The lower sensitivity in the modern case may be due to many factors, for example higher albedo due to the presence of ice sheets, and the higher percentage of land versus

ocean. The net incoming shortwave radiation from our late Eocene model has an extra 12 W/m$^2$ of insolation compared with the modern CM2.1, indicating a significantly lower albedo. Furthermore, our simulations indicate some state dependence, with higher sensitivity at higher CO$_2$.

The response of SST and SAT to CO$_2$ changes is shown in Figure 11. In the 1600 ppm case (Figure

11a,c), the tropics warm by between 3-4°C, with a strong amplification in both of the polar regions. SAT over the Arctic Ocean and over Antarctica are 7-9°C warmer, with the magnitude of polar amplification being similar in both hemispheres. This approximate doubling of warming in the polar regions aligns with expectations from idealised climate modelling of the polar amplification response (Alexeev et al., 2005). The polar amplification is a combination of the local radiative forcing and the

enhanced moist energy transport from the tropics. The Arctic Ocean warming is especially pronounced in the 1600 ppm case, despite the absence of ice-albedo feedbacks. Cloud radiation feedbacks may

provide additional polar amplification at high levels of $CO_2$ (Abbot et al., 2009). Warming over land shows a mixed response; warming over North America and Siberia is enhanced compared with oceanic regions at the same latitude, as is the case in southern Africa. By contrast, other regions such as Australia, equatorial Africa and parts of South America show little change to the land-ocean contrast.

This is linked directly with the land-ocean monsoon response in low to mid-latitudes. Regions of enhanced land warming are closely associated with enhanced drying, while regions of reduced land warming generally also become wetter, as shown in the P-E forcing (Figure 12). In the higher latitudes, the monsoonal forcing is weaker, and thus the P-E changes are not as closely linked with the temperature response.

In the 400 ppm case (Figure 11b,d), the magnitude of tropical cooling and polar amplification are similarly strong as in the warming case. The Antarctic SAT response is however somewhat stronger than the warming case due to the triggering of snow albedo feedbacks. Under 400 ppm, a far greater proportion of the Antarctic continent is covered in snow in winter. The model does not sustain freezing

temperatures over Antarctica in summer. However, this should not be interpreted directly as evidence that an ice sheet cannot be triggered under this climate; the model lacks the necessary feedbacks to simulate the accumulation of long term snow and ice (e.g. Gasson et al. 2014). One area of strong cooling occurs in the Weddell Sea, where SSTs drop by ~8°C, with a commensurate change in SAT. There is also a cessation of deep water formation in this region, and the formation of seasonal sea ice in

the 400 ppm case (see Figure 6). The Pacific sector of the Southern Ocean cools to a similar magnitude as the warming case. The SST cooling in the Arctic is less pronounced than in the warming case, simply because the sea surface freezes for much of the year and therefore the surface change is limited to ~4°C. The SAT response in the Arctic is similarly strong, showing that the atmosphere polar amplification can be much stronger than the Arctic SST change.

**4.2 Salinity and Hydrological Cycle**

In response to a doubling of $CO_2$ to 1600 ppm, surface salinity decreases markedly over the mid and high northern latitudes as shown in Figure 12a. Arctic Ocean salinities decrease by around 1 psu, due to

the enhancement of the high latitude precipitation and river runoff into the Arctic. The river runoff accumulates from North America, Europe and Siberia, so the net effect of the increased P-E (Figure 12c) on the salinity is particularly concentrated in the Arctic. This Arctic freshening also has a marked impact on North Atlantic salinity. Conversely when $CO_2$ is halved, Arctic salinities are enhanced by 1-2

psu, due to the weakening of precipitation and runoff into the Arctic.

Salinity changes in other regions also reflect changes in the P-E balance to a lesser extent. The north-east Pacific freshens in response to warming and becomes saltier in response to cooling, due to the same forcing mechanisms. The interiors of the gyre circulations in the Pacific show signatures of the P-E

forcing, but the salinity differences are not as clearly correlated here, due to advection and lateral mixing altering the signal. One area of notable increase in salinity under the warming scenario is the southern tropical Pacific, where salinity is greatly enhanced. This corresponds to an area of net drying in the P-E fields, but other regions do not show nearly the same salinity response to the same magnitude of forcing. Clearly advection feedbacks are at play, as evidenced also by the net freshening of the South

Atlantic in the 1600 ppm case. This occurs despite a net decrease in P-E over the subtropical South Atlantic gyre. One possible explanation is the enhanced African monsoon, which leads to large river runoff increases from West Africa into the Atlantic. Furthermore, sinking in the Atlantic sector of the Southern Ocean decreases markedly in the 1600 ppm case, which is evident in both the mixed layer depths and age tracers at intermediate depth (not shown). This reduction in deep water formation in turn

reduces the salt advection into the Weddell Sea, leading to freshening at the surface. In the 400 ppm case, the tropical South Atlantic becomes saltier, which is likely due to a decrease in river runoff from West Africa. However, the Weddell Sea becomes fresher in this case, due to a reduction of sinking in the South Atlantic and its associated salt advection.

Greenhouse warming enhances the hydrological cycle, due to the increased capacity of the atmosphere to hold water vapour. This effect is known as the Clausius-Clayperon relation. In the zonal mean, this effect has been shown to create a pattern of change where wet regions get wetter and dry regions get

drier. A simple scaling relation can be derived that relates the change (denoted by δ) in Precipitation minus Evaporation (P – E) to its original distribution (Held and Soden, 2006):

$$\delta(P - E) = \alpha \, \delta T \, (P - E),$$

where $\alpha$ is a constant and $\delta T$ is the change in temperature. This scaling relation has been shown to hold true over ocean regions in modern observations (Durack et al., 2012), but not over land where the dynamics of rainfall distribution are more complex (Greve et al., 2014). In line with the above scaling argument, the oceanic patterns of wet vs dry regions (Figure 4d) are enhanced in most ocean regions when $CO_2$ is doubled. The equatorial response of P-E has some exceptions to the scaling relation, as the western Pacific warm pool and equatorial cold tongue do not respond according to the same paradigm. However, we note that the equatorial Pacific is subject to a strong zonal asymmetry due to El Nino variability, and these patterns do not follow the scaling relation as cleanly as the latitudinal relationship even in modern climate models (Held and Soden, 2006).

The basin integrated forcing of P-E and river runoff upon the surface ocean is summarised in Figure 13. This illustrates the combined effects of P-E forcing and river runoff in creating much fresher Arctic conditions compared to present day. In comparing the control run (800 ppm) with the 1600 ppm and 400 ppm cases, the warmer simulations give rise to increased freshwater forcing in the North Pacific to a similar degree as in the Arctic. However, the Arctic Ocean has a much smaller surface area and restricted outflow and therefore the impacts on surface salinity of this forcing are strongest. The connection of Arctic surface waters into the Atlantic gives rise to large salinity changes in the North Atlantic, whereas the North Pacific salinity is less impacted by this change. Furthermore, the freshwater forcing is concentrated in the north-east of the basin, away from the deep water formation region in the north-west. Thus the North Pacific is able to maintain deep water formation despite the enhanced freshwater forcing.

Overall, these changes in salinity are not enough to alter the preferred regions of northern sinking. The paleogeography employed here robustly generates sinking in the North Pacific and in the Southern Ocean. In all cases, the northern cell is warmer and saltier, and the southern cell is colder and fresher

and thus is heavier in the abyss, as in the present day. The cooling-induced weakening of the hydrological cycle does bring the North Atlantic closer to a regime of deep water formation, but these changes of 1-2 psu are not enough to trigger sinking. If we hypothesise that NADW did form at the EOT (Borrelli et al., 2014), then other mechanisms are needed to first raise the control state salinities in the North Atlantic.

## 4.3 Meridional Overturning Circulation Response

Meridional overturning circulation is weaker in the 1600 ppm case, by around 10 Sv in the northern cell, and up to 15 Sv in the southern cell (Figure 14a). Although this is a substantial reduction, the overall structure remains similar to that seen in the control run (Figure 7), with sinking in the North Pacific and Southern Ocean, albeit with a lower magnitude. The reduction in magnitude may be partly due to the 400 and 800 ppm runs being further from equilibrium, and therefore the deep ocean is expelling more heat. Otherwise, the high $CO_2$ case has a lower meridional temperature gradient, which may alter the forcing of the interhemispheric MOC (Wolfe and Cessi, 2014). In the 400 ppm case, the overall magnitude of the MOC is similar in magnitude to the control, however there is a northward shift of the southern cell, which manifests as a positive change in the high southern latitudes in Figure 14b. As the climate cools, a stable halocline develops in the Weddell Sea which ceases to form deep/bottom water. In winter, however, the halocline supports the growth of seasonal sea ice in the region.

In the control 800 ppm simulation, the surface waters in the Ross Sea are slightly warmer and more saline than those in the Weddell Sea, but they have densities sufficiently close to allow both regions to induce deep convection in winter (Figure 15). With lowered temperatures in 400 ppm simulation, the non-linear dependence on temperature of sea water density will for the same temperature decrease cause a larger density increase of the warmer Ross Sea surface waters. Presumably, this feature combined with the salt-advection feedback enhances the surface density difference between the two regions to the point where deep convection ceases in the Weddell Sea. As this transition occurs a stable halocline develops, supporting the growth of seasonal sea ice in the Weddell Sea (Goosse and Zunz, 2014). However, the net sea ice formation is too weak in the 400 ppm simulation to produce any brine enriched

deep water, which is an important feature for the present-day dense-water formation around Antarctica. In the 1600 pm case, freshening occurs due to enhancement of the hydrological cycle, leading to cessation of Weddell Sea convection. Vigorous sinking occurs in the South Pacific in all cases, analogous to Ross Sea convection in the present day. This area is more predisposed to sinking due to a relative lack of river runoff ending up in that region, while the strong subpolar gyre circulation allows lower latitude water to maintain a salt feedback into the region. North Pacific surface waters are warmer and saltier than in the Southern Ocean sinking regions, and thus establish a deep water mass of comparable density to that of the Southern Ocean. However, the Southern Ocean water mass, having lower temperature, becomes the densest water mass in the abyss due to thermobaric effects.

## 5 Summary and Conclusion

This work presents a new configuration of the GFDL CM2.1 climate model, configured for the late Eocene. To our knowledge, the ~1° ocean resolution is the highest ocean resolution in a coupled model of the late Eocene to date, while the lower resolution atmosphere ensures computational efficiency for long timescale simulations. This represents a significant step forward in resolution and accuracy in the representation of late Eocene paleogeography. However, we note that a parallel study currently under review (Baatsen et al., 2018) has also developed a late Eocene (38 Ma) climate model at 1° resolution. Their model shows qualitatively similar climatologies to ours, albeit with slightly higher greenhouse gas concentrations. Under this configuration, important gateways known to impact the ocean circulation at the EOT are better represented than has been done previously. The model is spun up for 6500 years from relatively warm initial conditions. This leads to a quasi-equilibrium steady state in surface climate, while the deep ocean is still gradually cooling at a rate of less than 0.1°C per century. The model shows relatively high sensitivity to $CO_2$ forcing, and a warm control climate under 800 ppm $CO_2$. There is some agreement with SST proxies, although the model still does not capture the very low gradients from equator to pole implied by the available proxy records.

The model exhibits El Niño-like variability in the equatorial Pacific, though the western Pacific warm pool is larger and more persistent than present day. This creates a La Niña-like background state and

more robust east-west temperature difference. There are still major uncertainties and areas for future improvement in the model. These include possible improvements to the aerosols, vegetation, soil and river runoff schemes, which have been configured based on an opportunistic adaptation of available data. The sensitivity of these schemes has not been explored here, and may provide avenues for future improvement. For example, the contribution of land albedo, itself a function of vegetation and soil properties, may have important impacts on both the land temperatures and the global meridional temperature gradient.

We find that the model exhibits sinking in the North Pacific and the Southern Ocean, under all levels of $CO_2$. The southern water mass is colder and fresher and dominates the abyssal ocean, while the northern deep water mass is warmer and saltier, analogous to the present-day structure of North Atlantic Deep Water overlying Antarctic Bottom Water. The model is configured with a bottom-enhanced mixing scheme and a uniform background diffusivity. Sensitivity tests indicate that using a Bryan-Lewis diffusivity scheme, commonly used in paleoclimate models, yields largely the same stratification and sinking regions. The Arctic Ocean is very fresh, with typical surface salinities of 20 psu, in agreement with Eocene salinity proxies. The connection between this fresh water mass and the North Atlantic prohibits the formation of North Atlantic Deep Water, since North Atlantic salinities are around 25-30 psu in present day sinking regions. These results highlight the importance of using late Eocene paleogeography in modelling the EOT. Using present day geography as a control state would not capture the dramatic differences in salinity in the northern ocean basins from the present day, which in turn greatly alter the preferred regions of sinking.

In response to $CO_2$ forcing, North Atlantic salinity responds clearly to a strengthening of the hydrological cycle under warming, and weakening under cooling. The net effect is a decrease of 1 psu in the North Atlantic for a doubling of $CO_2$, and an increase of 1-2 psu for a halving of $CO_2$. This effect is not enough to trigger North Atlantic sinking in the model. But it does suggest that if the climate were closer to a North Atlantic sinking regime at the EOT due to factors not captured here, $CO_2$ cooling could provide a trigger. This model provides a platform for further sensitivity studies in altering aspects

of the paleogeography. Changes to the Southern Ocean gateways, Arctic gateways, and the imposition of an ice sheet have all been shown to have significant impacts on the ocean circulation and climate in previous modelling studies. Deepening of the Greenland-Scotland Ridge around the EOT has been suggested as a potential trigger of North Atlantic sinking (Abelson and Erez, 2017; Stärz et al., 2017). It is important to investigate such changes in a paleoclimate model with an accurate representation of late Eocene paleogeography, so that changes across the EOT can be referenced to circulation, temperature and salinity characteristics that are appropriate to that time period.

## Acknowledgments

This work was supported by the Bolin Centre for Climate Research, Research Areas 1 and 6 and the Swedish Research Council project 2016-03912. Numerical simulations were performed using resources provided by the Swedish National Infrastructure for Computing (SNIC) at NSC, Linköping. MB is supported by the Netherlands Earth System Science Centre (NESSC) and the Ministry of Education, Culture and Science (OCW), grant number 024.002.001. The authors thank Casimir de Lavergne and an anonymous reviewer for their constructive comments, which helped to improve the manuscript. Model data can be made available on request.

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

**List of Figures**

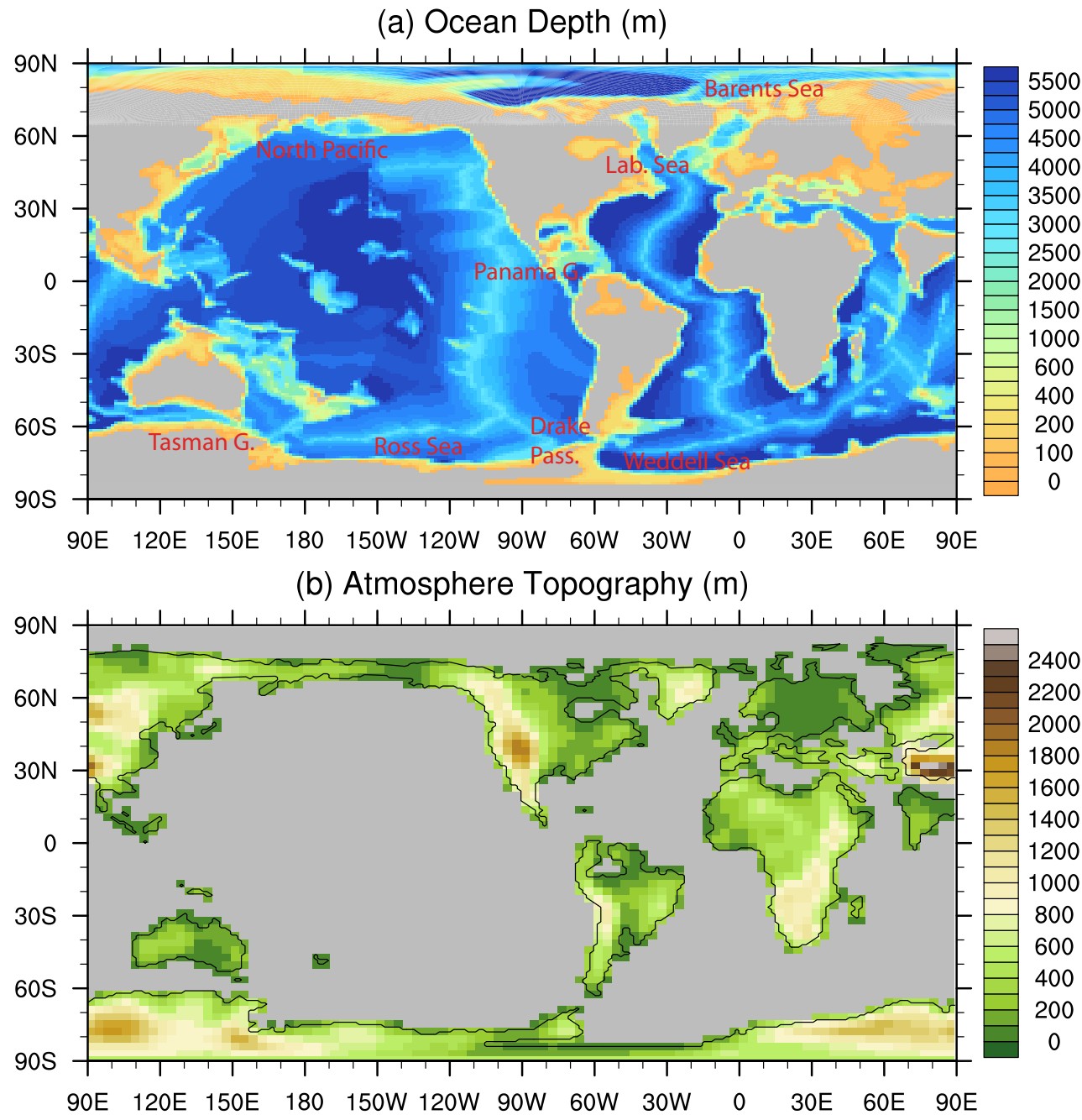

Figure 1: Bathymetry and topography of the late Eocene, adapted from (Baatsen et al., 2016). Both figures are plotted using a cell fill method, illustrating the resolution of the grid cells in each case. Due to the difference in resolution between the ocean and atmosphere, coastal grid cells in the atmosphere typically contain a fraction of both land and ocean.

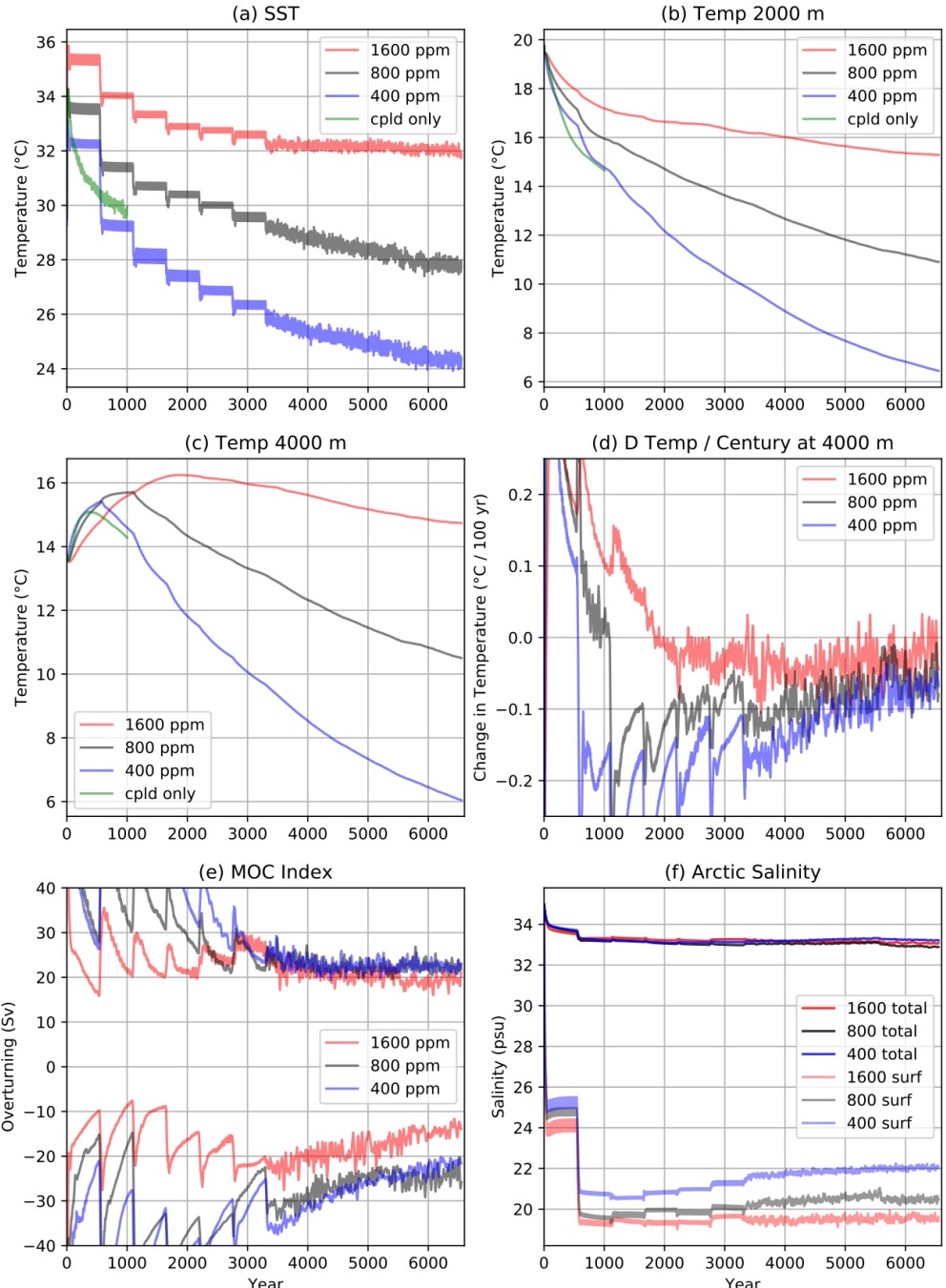

**Figure 2: Spinup evolution of the 400, 800 and 1600 ppm simulations, showing the global mean temperature evolution at (a) the surface, (b) 2000 m, (c) 4000 m and (d) the time rate of change at 4000 m. Plots (a-c) also include the 1000-year coupled-only run at 800 ppm, illustrating the faster evolution achieved in coupled mode. (e) Meridional overturning circulation (MOC) indices for each simulation: positive values indicate the maximum overturning value in the Northern Hemisphere, while negative values indicate the maximum overturning value in the Southern Hemisphere. (f) Arctic salinity evolution, showing the total (top three curves) and the surface average (bottom three curves). Plots (d) and (e) are filtered using a 21-year running mean.**

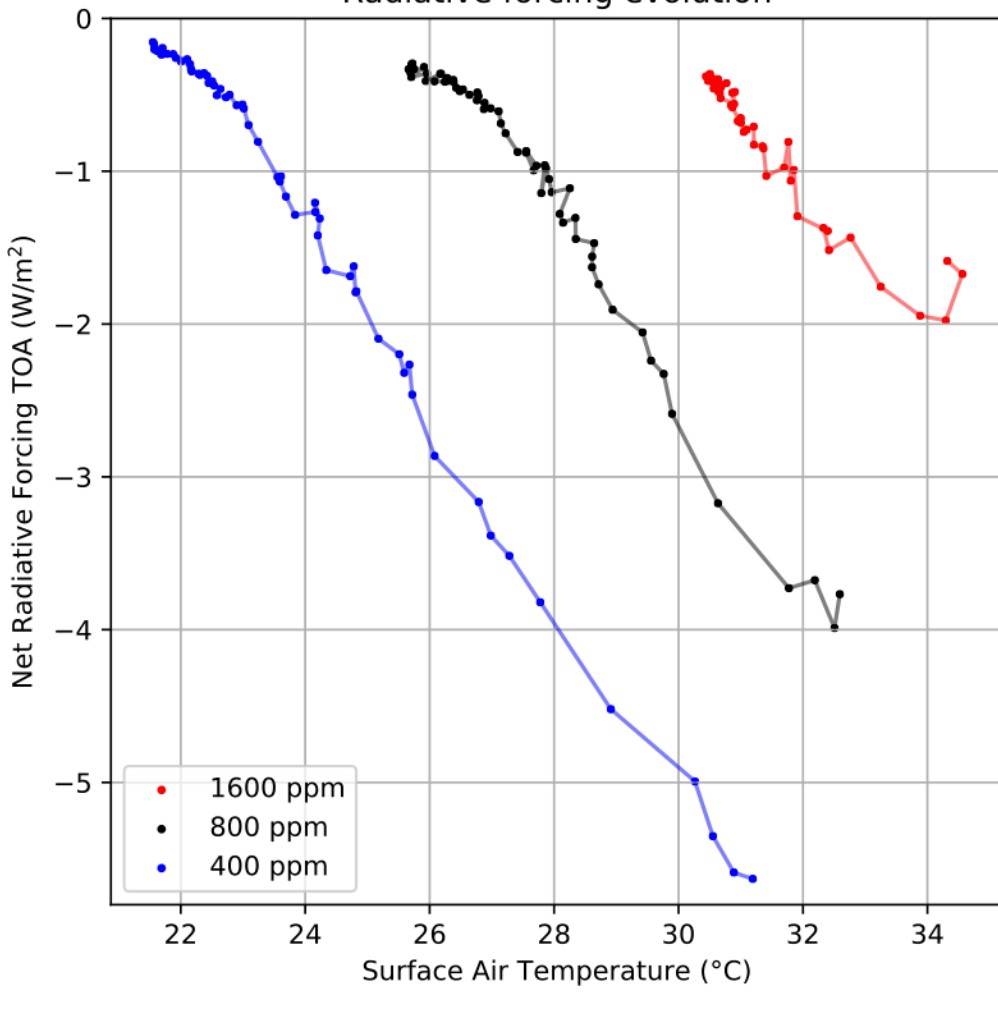

**Figure 3: Net radiative forcing at the top of atmosphere (TOA) plotted against surface air temperature. Data points are derived from each 10-year average during the iterative coupling phase of the spinup, and then from each 100-year average in the fully coupled phase. The first 10 years are omitted in all three cases since the net radiative forcing is temporarily positive. The time evolution goes from bottom-right to top-left in all cases (i.e. a cooling trend).**

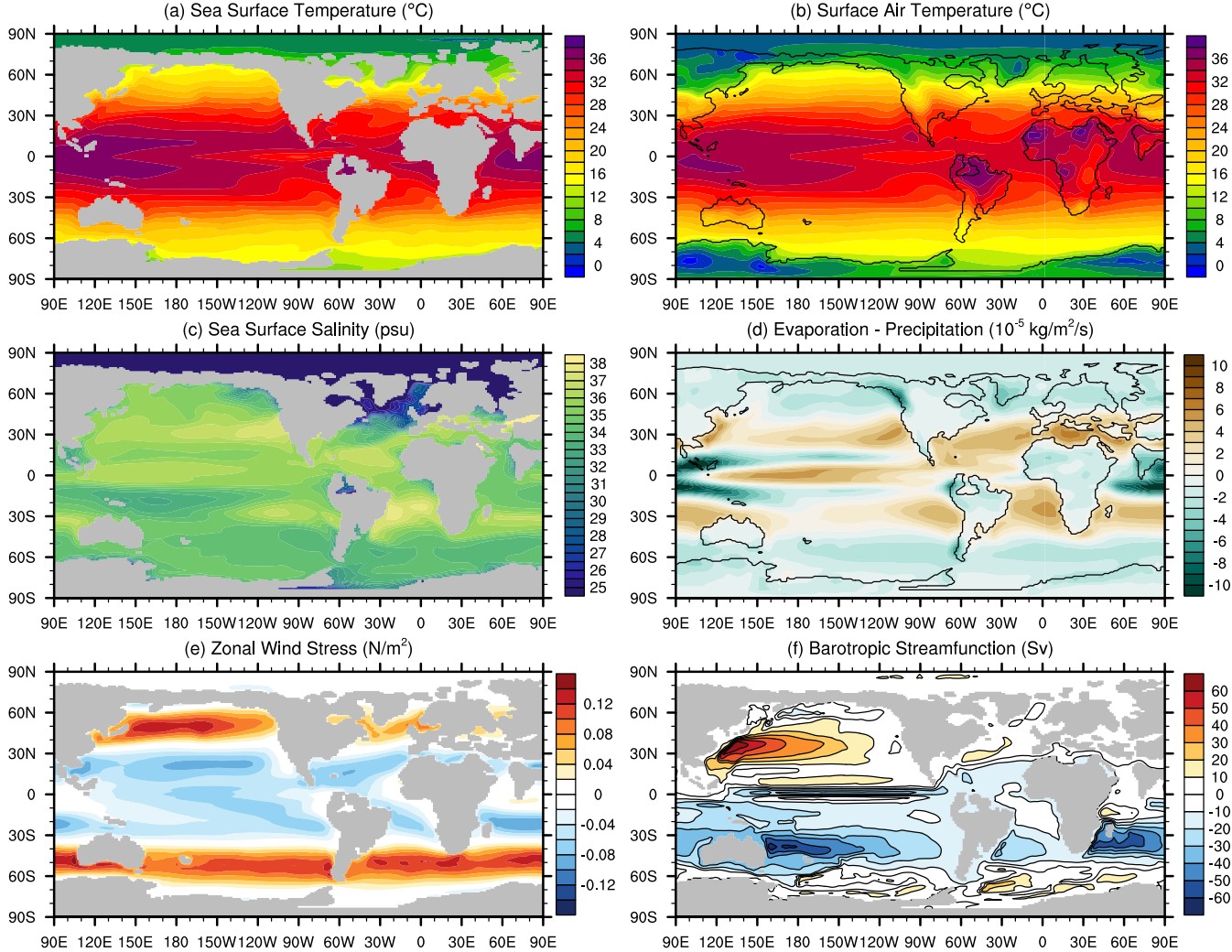

**Figure 4: Control climate state at 800 ppm, showing (a) sea surface temperature (SST), (b) surface air temperature (SAT), (c) sea surface salinity (SSS), (d) evaporation minus precipitation, (e) barotropic streamfunction and (f) zonal wind stress.**

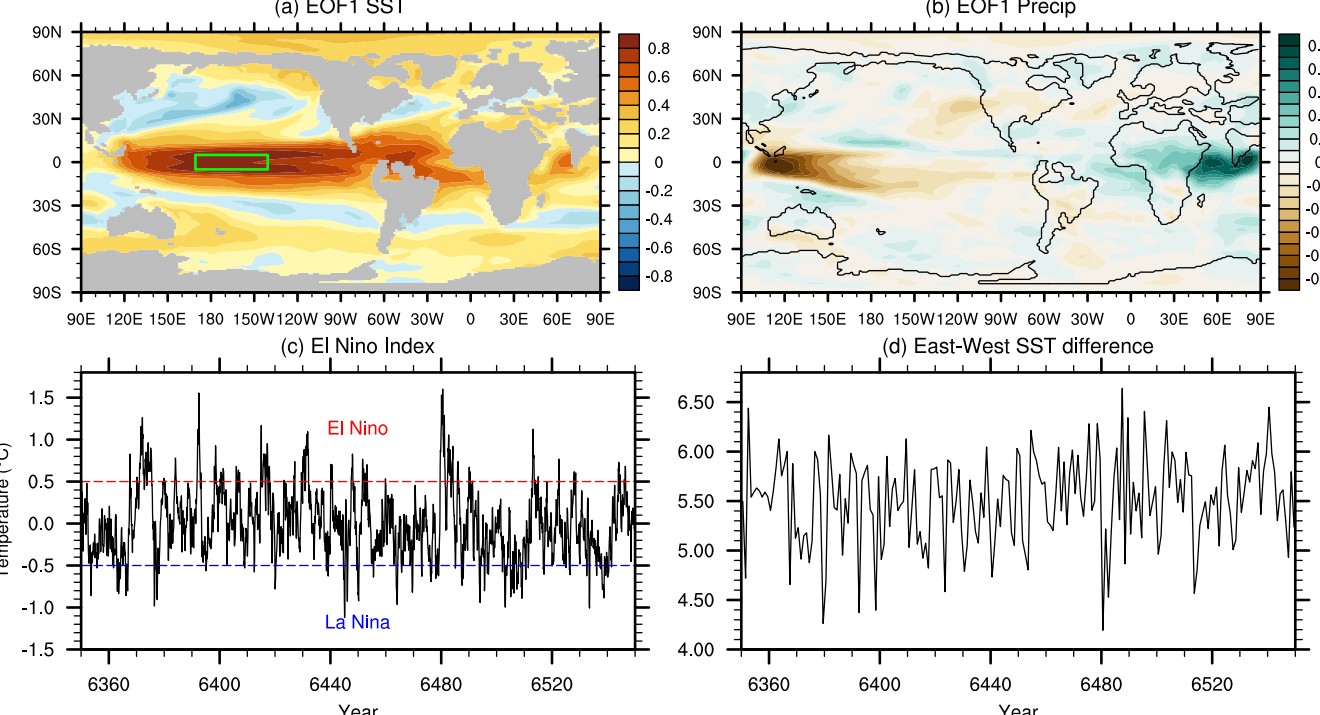

**Figure 5: (a) 1st Empirical Orthogonal Function (EOF1) of monthly sea surface temperature anomaly (SST), and (b) precipitation. From the EOF1 of SST we then define an El Niño index region from 170°E to 140°W, and from 5°S to 5°N, as shown in the green box in (a). This region was chosen as an 'El Niño index' to be representative of strongest variability in Pacific SST. (c) El Niño index variability in monthly SST anomaly. (d) Annual mean east-west Pacific SST difference.**

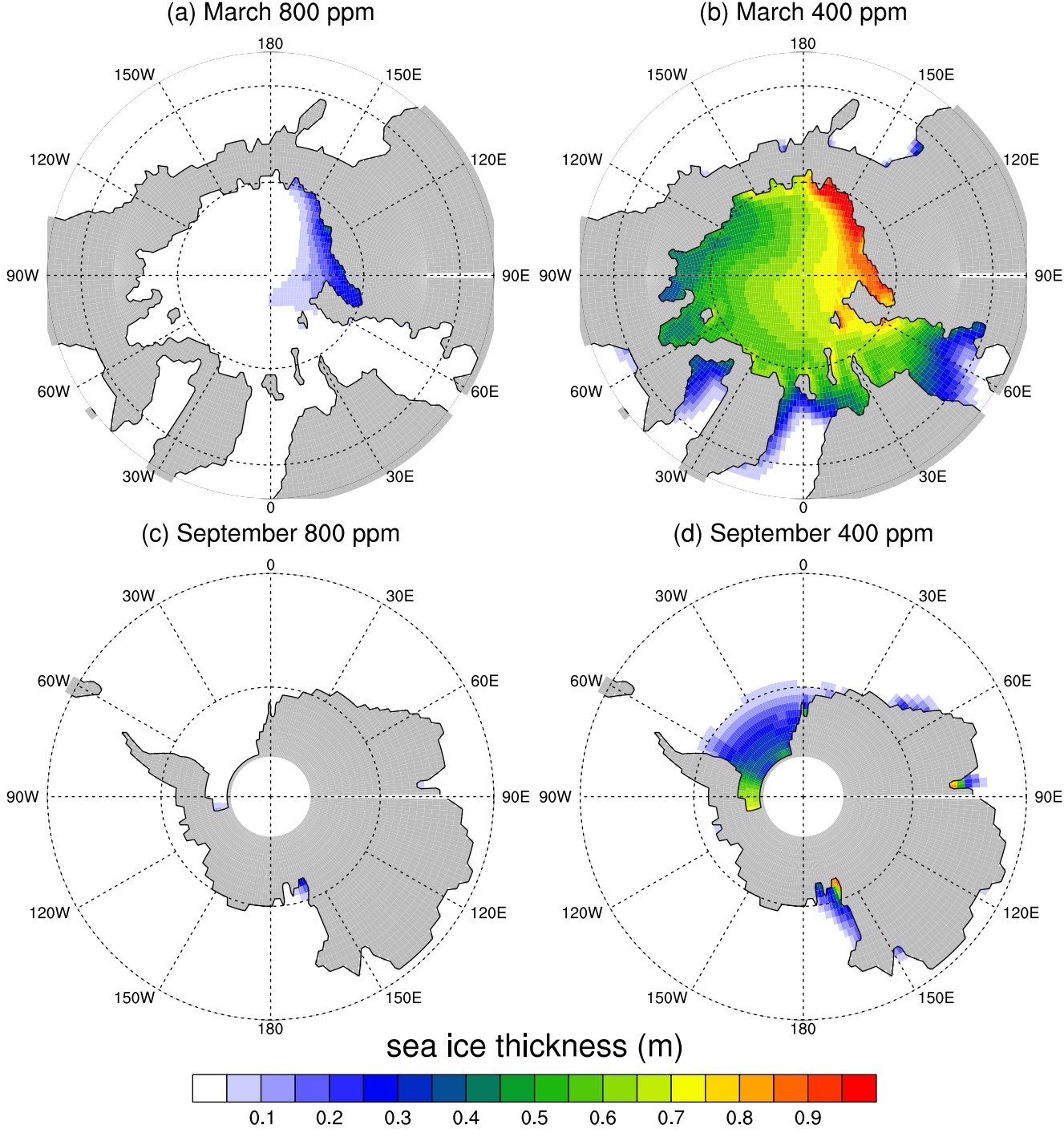

**Figure 6:** March (NH maximum) sea ice thickness of (a) the 800 ppm run, and (b) the 400 ppm run, and September (SH maximum) sea ice thickness for the (c) 800 ppm run and the (d) 400 ppm run. The 1600 ppm run is sea ice free all year.

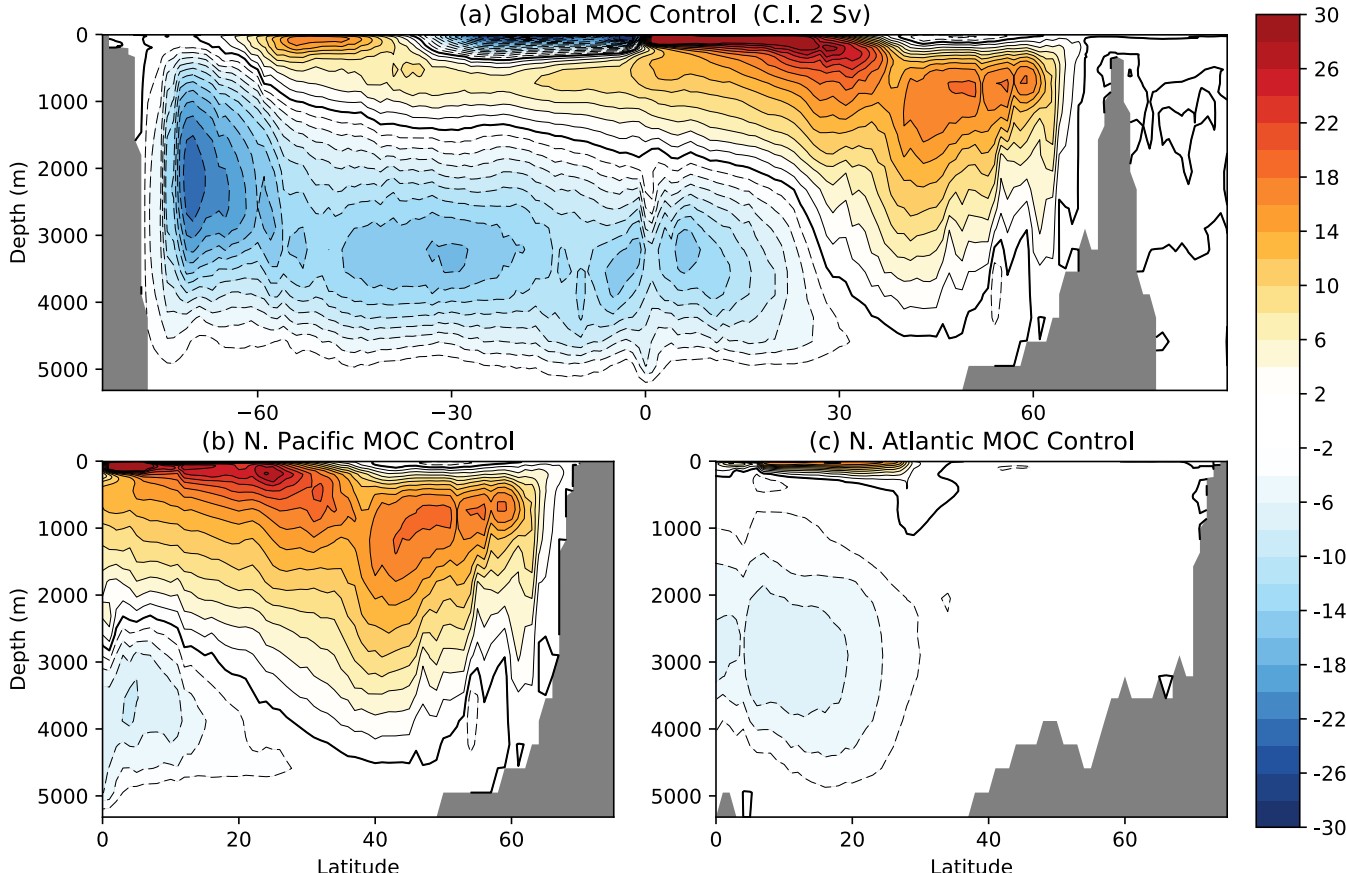

**Figure 7: (a) Global Meridional Overturning Circulation (MOC), with the northern hemisphere MOC split into (b) Pacific basin and (c) Atlantic basin.**

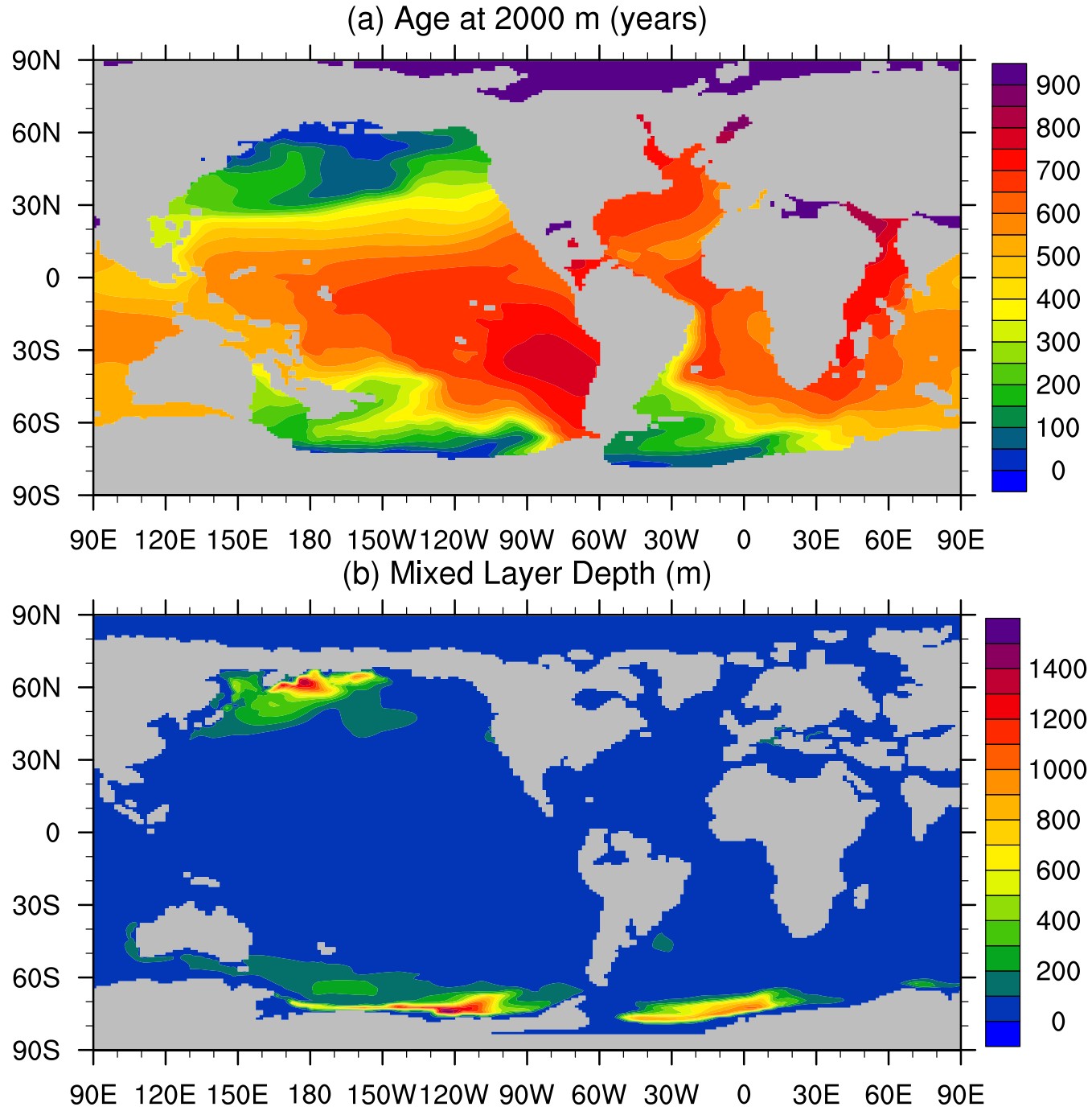

Figure 8: (a) Age tracer at 2000 m depth and (b) mixed layer depth. Note that the age scale is saturated in the Arctic, where the strong halocline prevents ventilation and ages are greater than 3000 years.

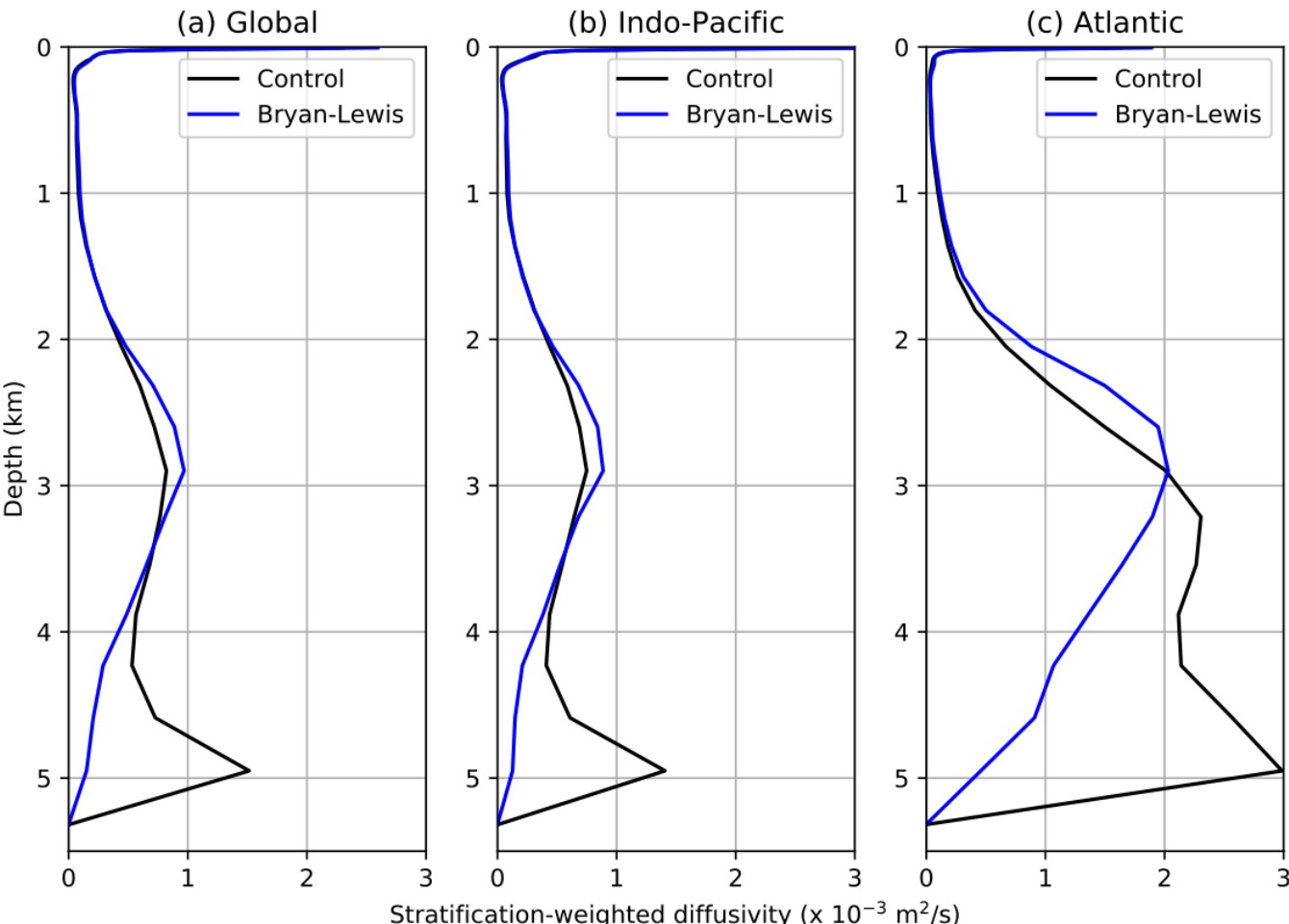

**Figure 9: Stratification-weighted diffusivity (see text) computed over (a) global, (b) Indo-Pacific and (c) Atlantic domains for the control (800 ppm) and Bryan-Lewis diffusivity experiments.**

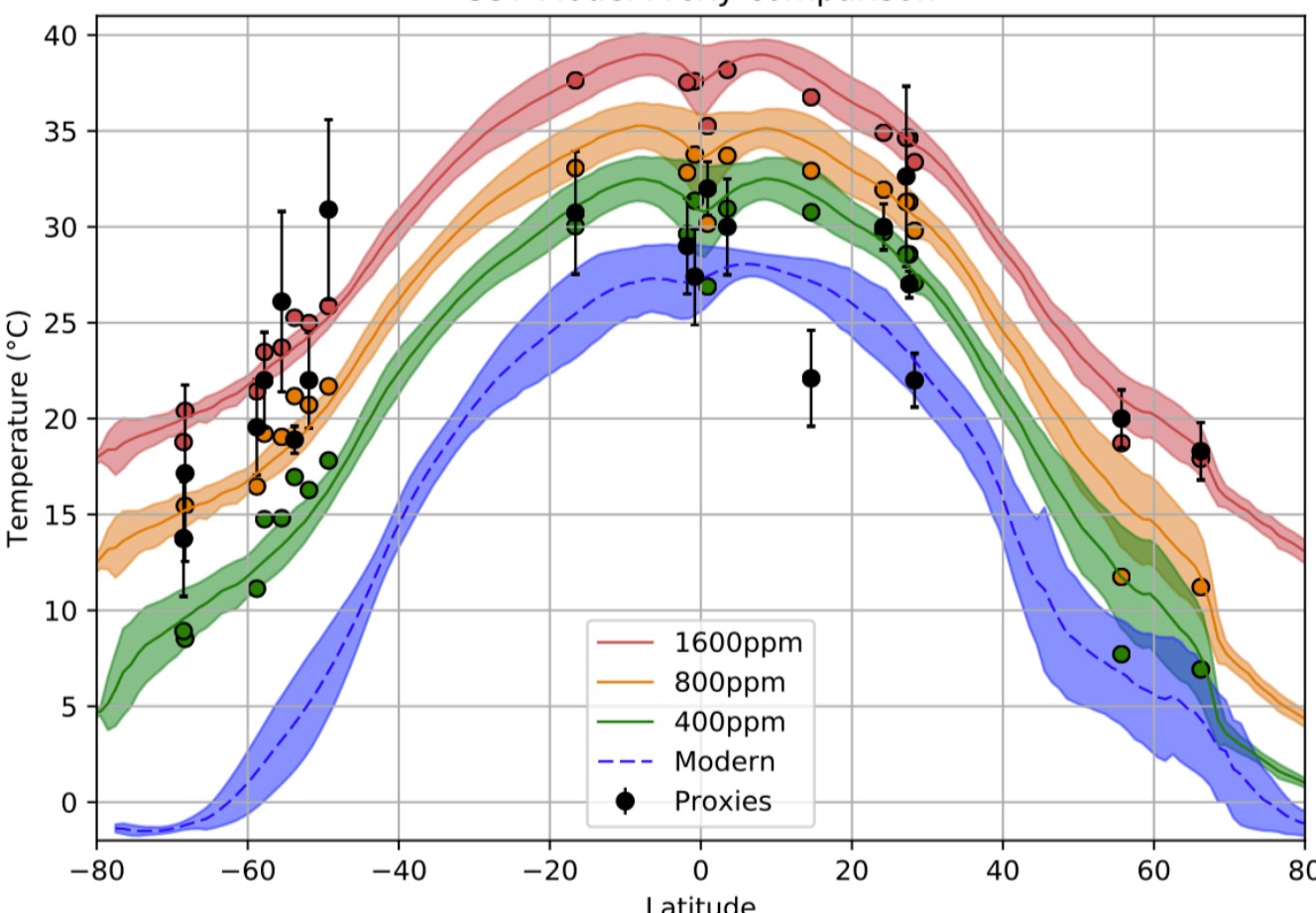

**Figure 10: Comparison of model 1600 ppm (red), 800 ppm (orange) and 400 ppm (green) simulations with a compilation of SST proxy data (black circles) from 38 to 34 Ma (Baatsen et al., 2018, Table A2). The compilation combines UK'$_{37}$ , TEX$_{86}$, Mg/Ca, $\delta^{18}$O and $\Delta_{47}$ data – see text for further references. The model temperature for each proxy location is shown by the coloured circles. The zonal mean and standard deviation are shown by the solid lines and shaded areas respectively. Also shown in blue is an equivalent distribution of zonal mean and standard deviation from the modern World Ocean Atlas (Locarnini et al., 2013).**

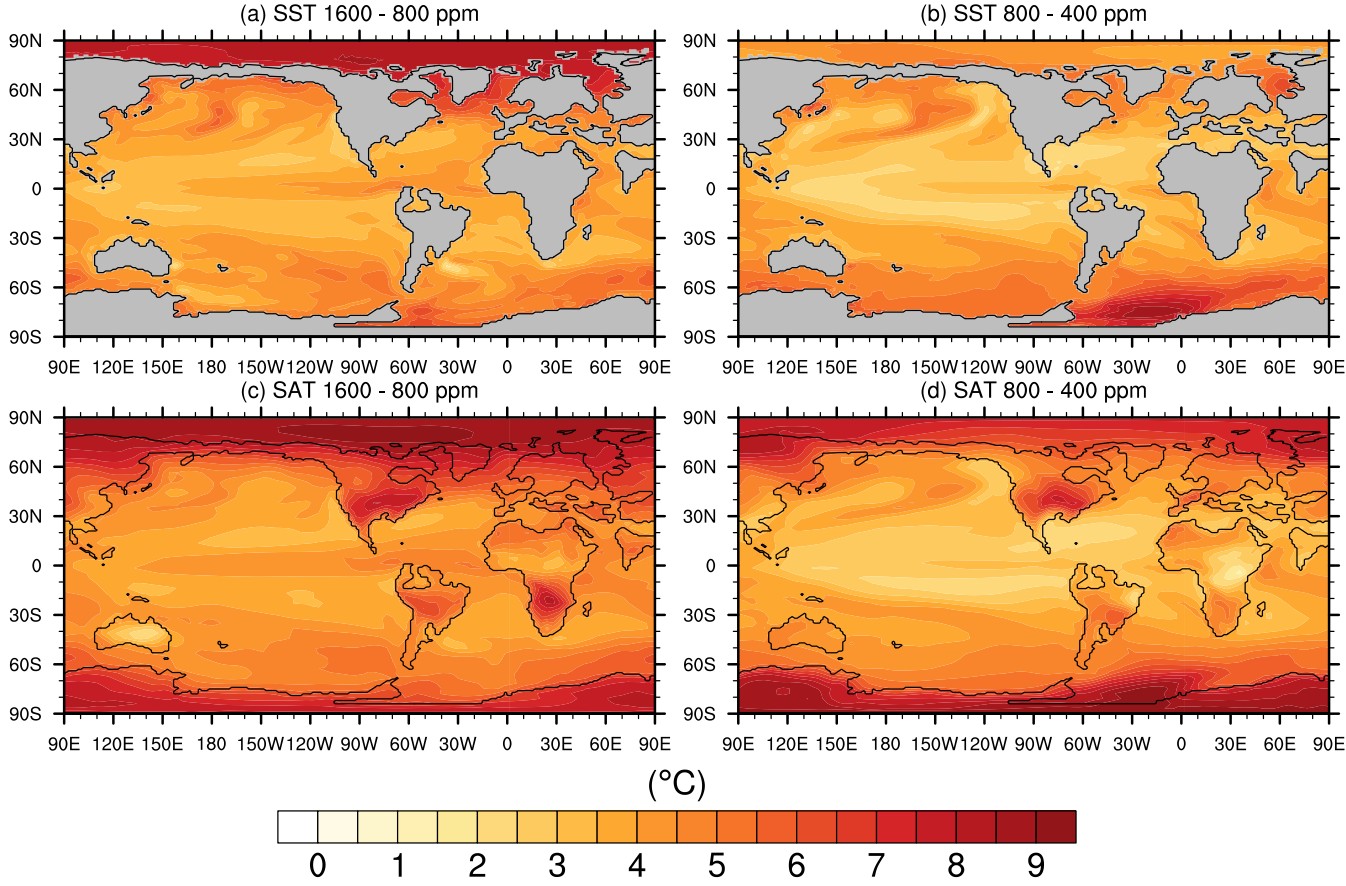

**Figure 11: Sea surface temperature (SST) difference between (a) 1600 – 800 ppm and (b) 800 – 400 ppm and surface air temperature (SAT) difference between (c) 1600 – 800 ppm and (d) 800 – 400 ppm.**

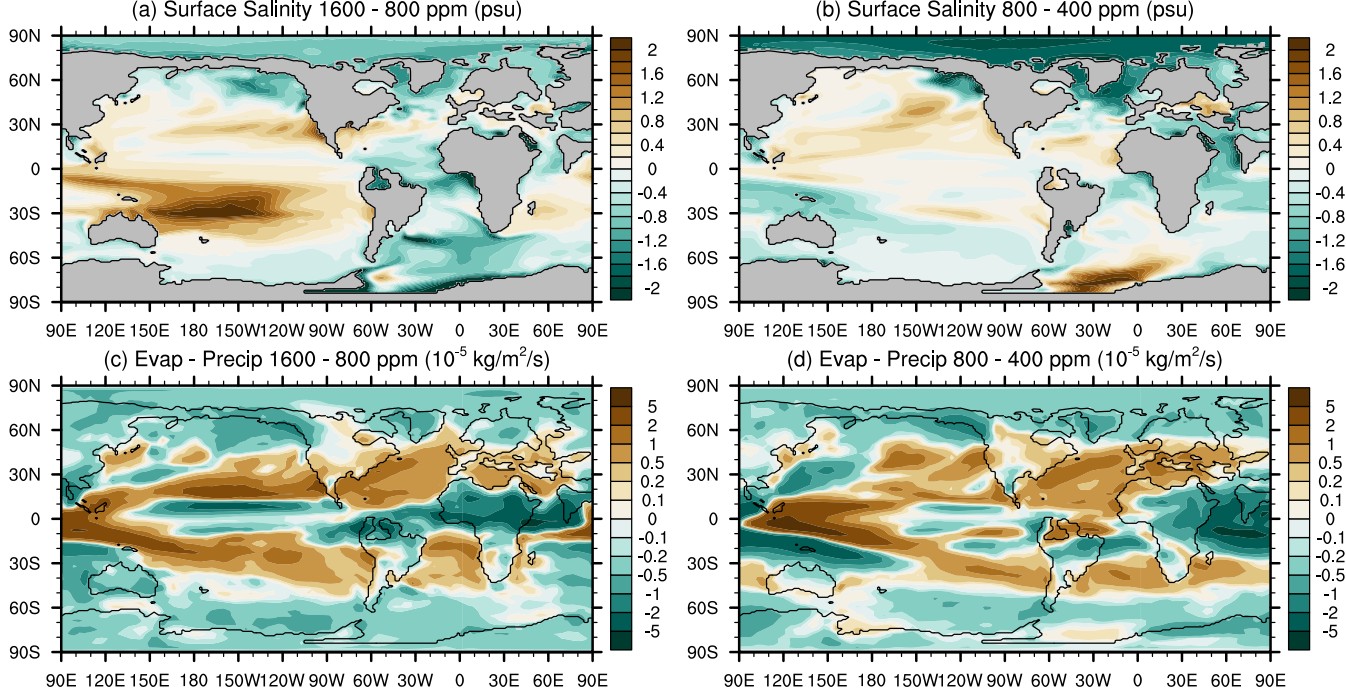

**Figure 12:** Surface salinity changes between (a) 1600 − 800 ppm and (b) 800 − 400 ppm, with the corresponding changes in evaporation minus precipitation for (c) 1600 − 800 ppm and (d) 800 − 400 ppm. Note the non-linear scale in plots (c-d).

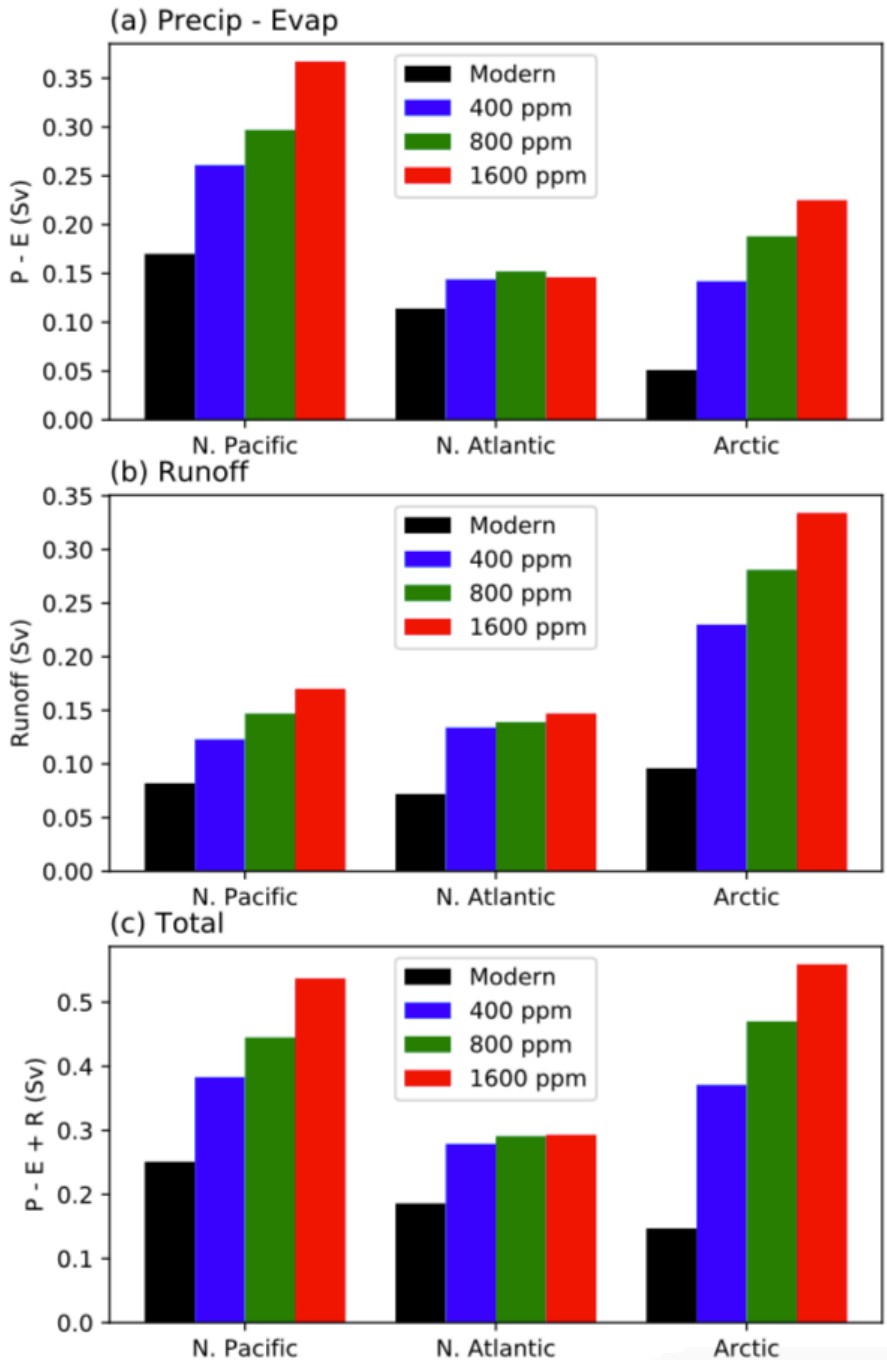

**Figure 13: (a) Precipitation – evaporation into the North Pacific and North Atlantic (both with a cutoff latitude of 45°N) and the Arctic Ocean, comparing a modern CM2.1 simulation with the late Eocene model at 400, 800 and 1600 ppm. (b) shows river runoff with the same breakdown as in (a), while (c) shows the total of precipitation – evaporation plus runoff.**

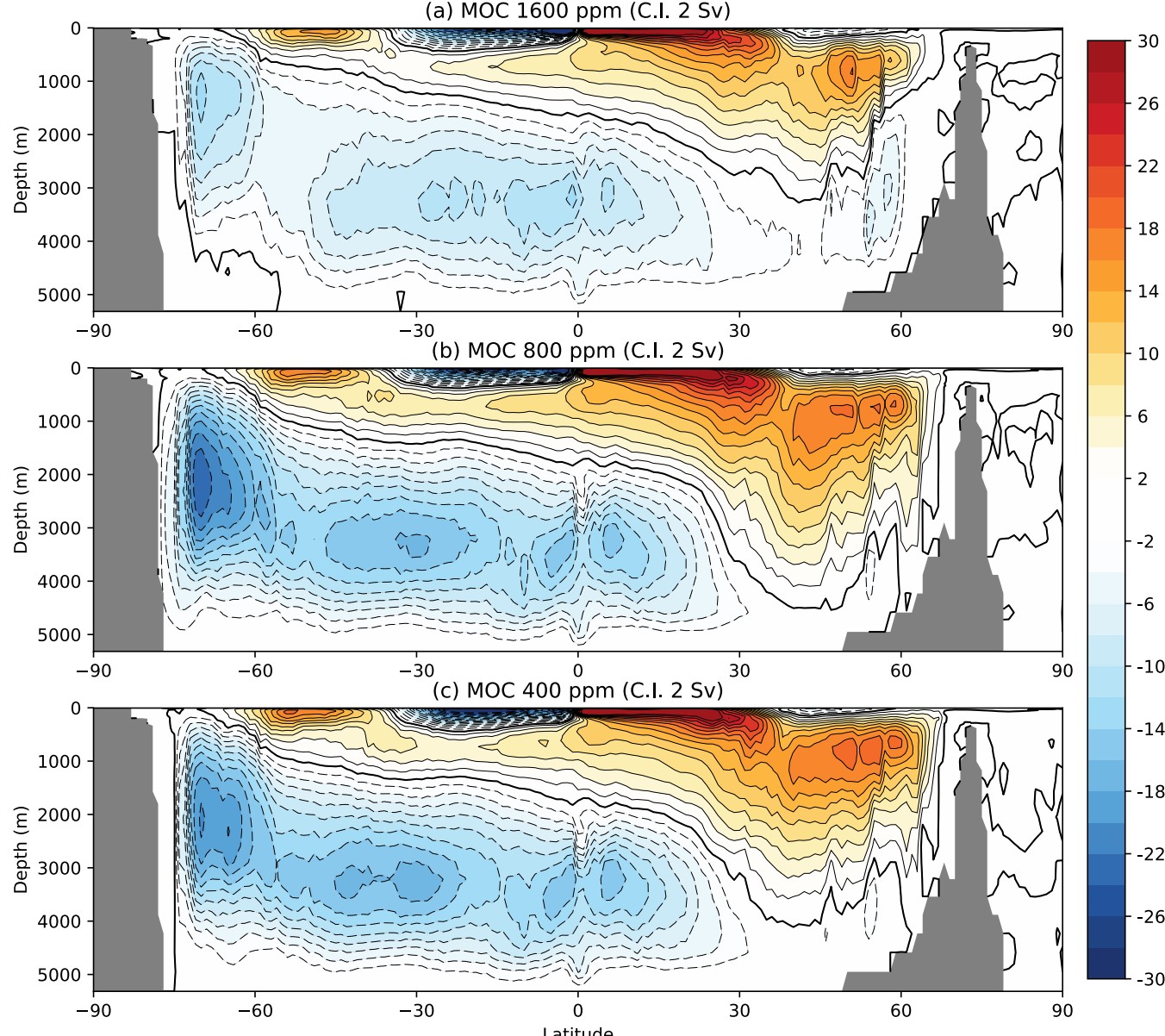

**Figure 14: Global meridional overturning circulation (MOC) for CO₂ levels of (a) 1600 ppm, (b) 800 ppm and (c) 400 ppm.**

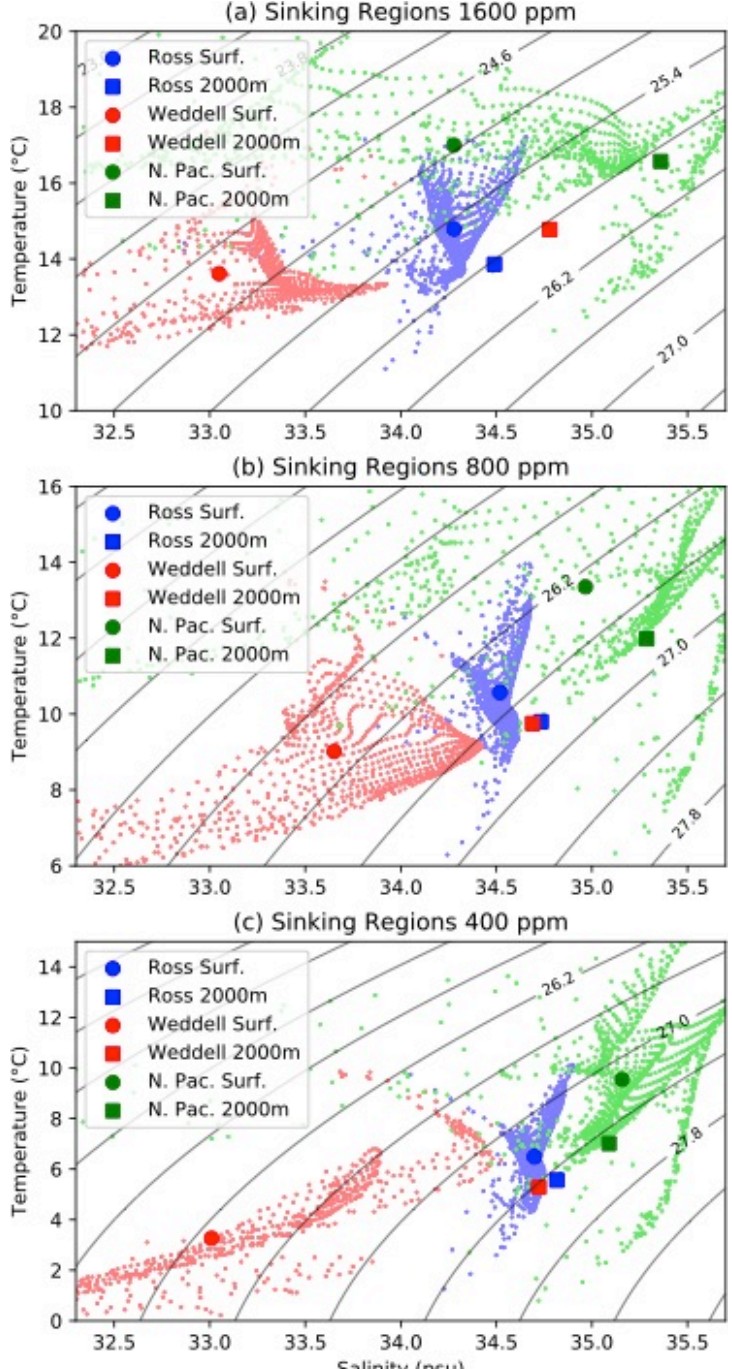

**Figure 15: Temperature and salinity properties in the sinking regions, showing surface values during winter months: July to September in the Southern Hemisphere and January to March in the Northern Hemisphere. Values are taken south of 60°S in the Southern Hemisphere and north of 50°N in the North Pacific. Shelf values less than 200 m depth are excluded. Bold circles indicate regional surface means, lighter dots indicate individual grid boxes, and bold squares indicate regional annual means at 2000 m. Contours of surface density are overlaid in each case.**

