# Peer review of "Climate sensitivity and meridional overturning circulation in the late Eocene using GFDL CM2.1"

_Climate of the Past, 2017_

## Referee Comment (RC1) · C. de Lavergne (Referee) · 8 Feb 2018

The authors describe a suite of four climate model simulations that use topography and boundary conditions representative of the Eocene-Oligocene transition ($\sim$ 34 Ma ago). The use of a recent reconstruction of the 34 Ma topography (Baatsen et al. 2016) and of relatively high resolution ($\sim$ 1° in the horizontal, 50 levels in the vertical) in the model ocean distinguishes these numerical experiments from previous attempts to model the late Eocene climate. The sensitivity of the simulated climate state to the prescribed level of atmospheric $CO_2$ (400 vs. 800 vs. 1600 ppm) and to the parameterization of ocean vertical mixing (bottom-enhanced mixing vs. Bryan-Lewis

diffusivity profile) are both examined. This represents a substantial modelling effort. The text and figures provide a clear overview of the simulated surface climate and deep ocean circulation as a function of CO2 and as compared with present-day climate. I therefore recommend publication. Specific comments and suggestions that may help to improve the manuscript are provided below.

Specific comments:

1. Link between ACC and NADW (p3, L10-12).

I believe it is not clear (from the literature) that the ACC favours NADW formation via mechanical mixing and Ekman upwelling. The latter occur in the absence of a significant ACC. Toggweiler and Samuels (1995) argued that the presence of a deep ACC may force the southward flow into the Southern Ocean, that compensates for the surface northward Ekman flow, to be relatively deep. Subsequent work, reviewed by Marshall and Speer (2012), showed that eddy-driven mass transports bypass the constraint identified by Toggweiler and Samuels. Elsworth et al. (2017) suggest that the impact of the ACC on NADW may occur via density decrease of AABW. The authors may want to clarify the mechanistic link between ACC and NADW.

2. Topography (p7, L11-14).

It would be useful to provide some more information about the methodology and uncertainties underlying the Baatsen et al (2016) topography used here, since this a crucial (distinguishing) ingredient of the simulations. For example, how well constrained are the sizes of the shallow Arctic-Atlantic and Arctic-Thethys gateways? You mention smoothing of the land topography using a 3-point mean filter: is this a standard procedure (and if so, is there a reference for it)?

3. Vertical mixing schemes (p8, L6-7; p14, L5-20).

A more detailed description of the two mixing schemes and of the diffusivities they prescribe/predict would be welcome. I do not think the Bryan-Lewis (BL) scheme is

"specifically tuned to modern observations of mixing": it is merely an ad hoc diffusivity profile, with a smaller value in the upper-ocean than in the deep ocean. It would be useful to mention the BL diffusivity values and the transition depth. To better understand why the two mixing schemes produce similar results, calculating/showing the stratification(N2)-weighted global (or basin) mean diffusivity profile would help. A map of the stratification-weighted vertical mean diffusivity would also help to visualize the heterogeneity/intensity of mixing, in particular within shallow regions when using the modified Simmons et al. (2004) scheme.

Does the bottom-enhanced mixing scheme produce similar effective diffusivities than BL in the thermocline and in the abyss? Is the poleward heat transport also insensitive to the mixing scheme? Could relatively high effective thermocline diffusivities contribute to the relatively warm climates simulated by this model configuration?

It should also be kept in mind (perhaps mentioned) that the Simmons et al. (2004) scheme assumes that the energy input to mixing within each grid column is proportional to the simulated bottom stratification. An increase in bottom stratification (as may result from increased high-latitude surface density gradients) is thus immediately paralleled by an increase in mixing energy, effectively maintaining roughly-constant diffusivities and roughly-constant circulation rates despite intensified density contrasts. This makes the Simmons et al. (2004) somewhat akin to the BL scheme, in that the diffusivity rather than the mixing energy is fixed. (Both schemes disallow a control of the energy consumed by mixing, making the interpretation of circulation rate sensitivities delicate.)

4. Sensitivity of Southern Ocean deep water formation to CO2 (p17, L7-8; p18, L10-11; p19, L25-27; p20, L1-5).

Deep convection in the Weddell Sea shows an interesting non-linear dependence on CO2, being stronger in the control state (800 ppm) than at 400 or 1600 ppm. This dependence merits more in-depth discussion.

Reduced convection and overturning at 1600 ppm is suggested to result from a lower

meridional temperature gradient. This is not clear (existing theoretical arguments for a MOC-strength dependence on meridional density gradients are debatable and do not apply to the abyssal overturning), and if true relies on the stratification-dependence of the specified bottom-enhanced diffusivities. Enhancement of the hydrological cycle and consequent surface freshening of high latitudes (Figure 10) seems a more plausible/direct cause.

The cause of the absence of Weddell Sea deep convection at 400 ppm is not discussed. It is only noted that reduced sinking contributes to a fresher surface in the Weddell Sea, and that seasonal sea ice forms. The fact that deep convection ceases despite a weaker hydrological cycle and colder atmospheric winter temperatures may appear as a paradox. In fact, this response could reflect a fundamental regime shift as surface winter temperatures fall below $0°C$. If temperatures remain well above freezing, winter cooling drives sustained open-ocean convection. When winter temperatures approach $0°C$, cooling barely elevates surface densities (the thermal expansion coefficient is small) and sea ice forms before thermal open-ocean convection can occur. The cycle of sea ice formation and melting then establishes the halocline (the rejected brine mixes over a relatively deep layer, whereas the melt accumulates within a thin surface layer), which further stabilizes the near-surface ice-ocean system (e.g. Goosse and Zunz 2014). Here, there is sufficient subsurface warmth that sea ice formation remains modest (e.g. Martinson 1990) and that haline coastal or open-ocean deep water formation does not replace the thermal open-ocean convection simulated at 800 ppm.

Technical corrections:

p1, L17: "We employ an ocean resolution of. . ." -> "The atmosphere and ocean horizontal resolutions are . . .., respectively."

p1, L19; p8, L4: "simulate the model" -> "run the model".

p1, L20: "CO2" -> "atmospheric CO2".

p1, L26: "salinities" -> "surface salinities".

p2, L10: "role of the ocean circulation" -> "the role of ocean circulation".

p2, L11: "leading theories of what caused" -> "prevailing proposed mechanisms for". Should albedo feedbacks be mentioned as essential to these mechanisms?

p2, L16: "atmospheric CO2" -> "atmospheric CO2 (thereafter referred to as CO2)".

p3, L18: Delete "data".

p5, L3: "it does not capture the subsequent" -> "early and late Eocene topographies differ significantly due to".

p5, L19: "a necessity of" -> "dictated by".

p6, L12: "future perturbations" is somewhat unclear/amibguous. I would delete the unnecessary second half of the sentence ("and any.... late Eocene").

p6, L15: "the vertical diffusivity" -> "parameterized oceanic vertical mixing".

p6, L16-17: "mixing is set to a constant background value and then enhanced in the vicinity of rough-topography" -> "the diffusivity is set to a constant background value and enhanced near the bottom".

p8, L6-8: "since the BL scheme is..." -> "since the former is more physical".

p8, L8-10: Delete this last sentence of the paragraph, which is somewhat misleading.

p10, L14: Delete "similarly".

p11, L4: "cannot" -> "does not". Here and elsewhere, it is argued that salinities are too low for convection to occur. More cautious statements recognizing that what matters is the low salinity relative to that of other high-latitude seas (and hence relative to the deep ocean) would be preferable.

p11, L7: "forces" is perhaps too strong, given that other factors (e.g. Nilsson et al.

2013) also play a role?

p12, L12-14: Not clear to me, please explain why.

p13, L21-22: Delete "Both of these sectors. . .tracer distribution."

p13, L22: "The shallow gateways. . .provide barriers": rephrase.

p14, L1-2: It is not sufficient that the cell is "warmer and saltier". What matters is the temperature contrast traversed by the circulation.

p14, L3: "due to warm, salty water masses": unclear.

p14, L6-7: Delete "The shape of. . . modern geography".

p14, L8-10: Rephrase.

p14, L14: "roughness" -> "bottom-enhanced". Roughness is constant in your implementation. . .

p14, L15: "bottom roughness" -> "bottom-enhanced".

p14, L 15-16: "a larger separation. . . . theoretical predictions" -> "a weaker diffusive heat penetration into the abyss".

p15, L16: "apsects" -> "aspects". p16, L5: "for example higher albedo": not obvious, given potentially stronger albedo feedbacks.

p19, L7: "clear" -> "immediate".

p20, L17: "The" -> "the".

p22, L4: "bottom roughness" -> "bottom-enhanced".

p22, L8: "freshwater" -> "fresh water".

p22, L9-10: "even in the absence of tectonic barriers in the Nordic seas": what do you mean here?

[Figure]

Figures 6 and 12: Showing the MOC of the three (or four) runs in the same figure, in absolute values rather than as differences, would be much clearer.

---

## Referee Comment (RC2) · Anonymous Referee #2 · 26 Feb 2018

First of all, I did not read the other online review to this paper prior to writing my own, so this review is completely independent.

This paper aims to explore climate sensitivity and ocean overturning at the Eocene-Oligocene, as well as other aspects of the Earth system. It argues that by using more realistic paleogeographies and a relatively high resolution ocean model, it represents an advancement over previous work.

In essence, it describes a set of simulation at 3 $CO_2$ levels for the late Eocene, with 2 different parameterisations of ocean mixing, and includes a model-data comparison.

Overall, the paper provides an interesting summary of the basic state and climate sensitivity of this particular model, and is a solid baseline from which additional sensitivity studies can be carried out.

**Major comments:**

The introduction is a nice overview of some previous modelling work. However, more quantitative information could be presented. This paper focusses mostly on climate sensitivity and overturning, so a summary table of e.g. climate sensitivities and ocean overturning states at different $CO_2$ levels in previous work would be very informative and provide more motivation and context for the work presented here (especially if the paleogeogs and ocean resolution were included in the same table).

Section 2 on the model is good, but it appears that several changes have been made (to e.g. the resolution) compared with the published version of the model. As such, it is important to present an assessment of the performance of the model under preindustrial conditions. In addition, at several points in the manuscript comparisons are made with modern, and these are rather meaningless unless they are comparisons with a modern model simulation from the same model.

The spinup plots are useful and interesting, but it would be very useful to see Gregory plots of the three simulations as well, to assess how spun up the simulations are, and the likely equilibration temperature of the simulations.

**Specific Comments:**

Abstract: Is the model resolution really greater than previous work? A summarising table of previous simulations would help, see above.

P1, Line 25; P22, line 1: what is meant by "robust" in this context?

P2, Line 14: for a review of the mechanisms see Lear (2016, Science)

P2, line 21: seasonality as well as climate sensitivity was important in the Gasson paper.

P2, line 22: make it clear whether $CO_2$ estimates are for just prior to, or just after the EOT.

P6, line 10: need to make it clear that although using "realistic" paleotopographies is a step forward compared with some previous studies, there are still uncertainties in these that are not captured in this study.

P9, your experiences with model spinup are very interesting and I think warrant inclusion in the conclusions section, and maybe even the abstract.

P9, line 15: a graph illustrating the fully coupled versus asynchronous spinup procedure (e.g. a graph of time against global mean temperature at a couple of ocean depths) would be very interesting and illuminating. i.e. add non-asynchronous run to Figure 2.

P9, line 23: add another plot which is the evolution of ocean overturning strength in each basin over the model spinup period.

P9, line 23: Also add another timeseries which is ocean salinity over time in the Arctic (and maybe other ocean basins). It would be interesting to know if this is still decreasing or whether it has reached an equilibrium.

P11, line 27; p21, line 19: "La Nina-like mean state" I would avoid this term; it has caused much confusion in the Pliocene community. Instead, just say that in the annual mean the west-east gradient in increased.

P12, line 26: I don't think that the level of confidence in the interpretation of the proxy data justify this statement. I don't think the data really allows us to say which modelled seaice distribution is best (i.e. there is some seaice at 800ppmv).

P14, line 5: It would be good to note whether these sensitivity studies to mixing scheme were run for exactly the same length of time and had the same spinup process as the 'standard' runs.

P16, line 4 – for the comparison with modern climate sensitivity, is this exactly the same model and same resolution etc?

P16, line 16: "We suggest that the ice-free conditions allow for substantial radiative warming of SST in addition to atmospheric polar amplification." I don't understand what this means.

Section 4.4: This section is somewhat superfluous, and doesn't add much to the manuscript unless the location of late Eocene palms are plotted on Figure 13.

**Technical comments:**

Figure 1a – get rid of the 'hole' at the north pole and 0 degrees east.

Figure 3 caption – state the co2 concentration.

Coloured circles in figure 8 are difficult to see – put a black line around the circumference.

P15, section 4 title – $CO_2$ not CO2.

Figure 9 would be better presented as both plots being a warming, i.e. 1600-800 and 800-400. In fact, throughout the paper I would prefer presenting the simulations as 3 in a series, rather than one in the middle with a warmer and a cooler either side, to aid consistency.

---

## Author Comment (AC1) · 9 May 2018

We thank the reviewer for his thoughtful and constructive comments, which have helped to improve the manuscript. We present the reviewer's comments in black text, and our response in blue text.

**Summary:**

The authors describe a suite of four climate model simulations that use topography and boundary conditions representative of the Eocene-Oligocene transition (âĹij 34 Ma ago). The use of a recent reconstruction of the 34 Ma topography (Baatsen et

al. 2016) and of relatively high resolution ($\sim 1°$ in the horizontal, 50 levels in the vertical) in the model ocean distinguishes these numerical experiments from previous attempts to model the late Eocene climate. The sensitivity of the simulated climate state to the prescribed level of atmospheric $CO_2$ (400 vs. 800 vs. 1600 ppm) and to the parameterization of ocean vertical mixing (bottom-enhanced mixing vs. Bryan-Lewis diffusivity profile) are both examined. This represents a substantial modelling effort. The text and figures provide a clear overview of the simulated surface climate and deep ocean circulation as a function of CO2 and as compared with present-day climate. I therefore recommend publication. Specific comments and suggestions that may help to improve the manuscript are provided below.

We thank the reviewer for the overall positive assessment.

**Specific comments:**

**1. Link between ACC and NADW (p3, L10-12).**

I believe it is not clear (from the literature) that the ACC favours NADW formation via mechanical mixing and Ekman upwelling. The latter occur in the absence of a significant ACC. Toggweiler and Samuels (1995) argued that the presence of a deep ACC may force the southward flow into the Southern Ocean, that compensates for the surface northward Ekman flow, to be relatively deep. Subsequent work, reviewed by Marshall and Speer (2012), showed that eddy-driven mass transports bypass the constraint identified by Toggweiler and Samuels. Elsworth et al. (2017) suggest that the impact of the ACC on NADW may occur via density decrease of AABW. The authors may want to clarify the mechanistic link between ACC and NADW.

We have clarified the text in this section to say: "This notion rests on the modern-climate interpretation that, in addition to diapycnal mixing in the interior ocean (Munk and Wunsch, 1998), wind-driven upwelling in the ACC is a major driver of the deepwater cell presently associated with the Atlantic MOC (Toggweiler and Samuels, 1995). However, transports by mesoscale eddies counter the wind-driven upwelling in the

ACC, reducing the link between the winds over the ACC and the strength of the MOC (Marshall and Speer, 2012)."

**2. Topography (p7, L11-14).**

It would be useful to provide some more information about the methodology and uncertainties underlying the Baatsen et al (2016) topography used here, since this a crucial (distinguishing) ingredient of the simulations. For example, how well constrained are the sizes of the shallow Arctic-Atlantic and Arctic-Tethys gateways? You mention smoothing of the land topography using a 3-point mean filter: is this a standard procedure (and if so, is there a reference for it)?

The Baatsen et al (2016) topography is distinct from previous reconstructions (e.g. Markwick, 2007) in that it uses a Paleomagnetic reference frame to position the continents, rather than a hotspot reference frame. The other feature is that it starts from ETOPO modern day topography on land and relocates the continents by plate tectonic motion. For the deep ocean, an age-depth relationship is applied from Müller et al (2008), and adjusted to the Paleomagnetic reference frame.

Manual adjustments are then applied to areas where elevation changes are well constrained by geological evidence. Specific regions of adjustment include Antarctica, the Himalayas, the Amazon, Turgai Strait and the Tethys Sea. Perhaps the biggest drawback of this method is that it defaults too much to modern elevation. I.e. where paleo-elevation data was either missing or unknown to the authors at the time of publication, the ETOPO data fills in the spaces. We have added text to clarify this point. We also discuss alternative reconstructions of the Arctic-Atlantic and Arctic-Tethys gateways.

The 3-point mean filter was an ad-hoc adjustment, due to numerical noise in the atmospheric velocities that we encountered in the high-latitudes when first testing the model. This noise was most likely due to topographic variations on Antarctica, which needed to be smoothed out due to convergence of meridians on the topography grid. In hindsight the smoothing may not have been necessary other than on Antarctica. We
acknowledge that this representation of atmosphere topography could be improved.

**3. Vertical mixing schemes (p8, L6-7; p14, L5-20).**

A more detailed description of the two mixing schemes and of the diffusivities they prescribe/predict would be welcome. I do not think the Bryan-Lewis (BL) scheme is "specifically tuned to modern observations of mixing": it is merely an ad hoc diffusivity profile, with a smaller value in the upper-ocean than in the deep ocean. It would be useful to mention the BL diffusivity values and the transition depth. To better understand why the two mixing schemes produce similar results, calculating/showing the stratification(N2)-weighted global (or basin) mean diffusivity profile would help. A map of the stratification-weighted vertical mean diffusivity would also help to visualize the heterogeneity/intensity of mixing, in particular within shallow regions when using the modified Simmons et al. (2004) scheme.

Does the bottom-enhanced mixing scheme produce similar effective diffusivities than BL in the thermocline and in the abyss? Is the poleward heat transport also insensitive to the mixing scheme? Could relatively high effective thermocline diffusivities contribute to the relatively warm climates simulated by this model configuration?

It should also be kept in mind (perhaps mentioned) that the Simmons et al. (2004) scheme assumes that the energy input to mixing within each grid column is proportional to the simulated bottom stratification. An increase in bottom stratification (as may result from increased high-latitude surface density gradients) is thus immediately paralleled by an increase in mixing energy, effectively maintaining roughly-constant diffusivities and roughly-constant circulation rates despite intensified density contrasts. This makes the Simmons et al. (2004) somewhat akin to the BL scheme, in that the diffusivity rather than the mixing energy is fixed. (Both schemes disallow a control of the energy consumed by mixing, making the interpretation of circulation rate sensitivities delicate.)

We agree that the Bryan-Lewis mixing scheme is more accurately described as an ad-hoc diffusivity, with stronger mixing in the abyss than in the upper ocean. We have

added the diffusivity values and the transition depth to the manuscript in order to more properly describe the Bryan-Lewis scheme. In addition, we will compute stratification-weighted diffusivity values for each mixing scheme, in order to provide an 'effective diffusivity' as suggested. We have also added a basin-by-basin estimate of effective diffusivity. We do find that the effective mixing through the thermocline is similar between the two schemes, and this does help to explain the similarity in circulation.

We have also clarified our description of the Simmons et al (2004) mixing scheme, and its drawbacks associated with energy constraints, as suggested above.

**4. Sensitivity of Southern Ocean deep water formation to CO2 (p17, L7-8; p18, L10-11; p19, L25-27; p20, L1-5).**

Deep convection in the Weddell Sea shows an interesting non-linear dependence on $CO_2$, being stronger in the control state (800 ppm) than at 400 or 1600 ppm. This dependence merits more in-depth discussion.

Reduced convection and overturning at 1600 ppm is suggested to result from a lower meridional temperature gradient. This is not clear (existing theoretical arguments for a MOC-strength dependence on meridional density gradients are debatable and do not apply to the abyssal overturning), and if true relies on the stratification-dependence of the specified bottom-enhanced diffusivities. Enhancement of the hydrological cycle and consequent surface freshening of high latitudes (Figure 10) seems a more plausible/direct cause.

The cause of the absence of Weddell Sea deep convection at 400 ppm is not discussed. It is only noted that reduced sinking contributes to a fresher surface in the Weddell Sea, and that seasonal sea ice forms. The fact that deep convection ceases despite a weaker hydrological cycle and colder atmospheric winter temperatures may appear as a paradox. In fact, this response could reflect a fundamental regime shift as surface winter temperatures fall below $0°C$. If temperatures remain well above freezing, winter cooling drives sustained open-ocean convection. When winter temperatures

approach 0âŮęC, cooling barely elevates surface densities (the thermal expansion co-efficient is small) and sea ice forms before thermal open-ocean convection can occur. The cycle of sea ice formation and melting then establishes the halocline (the rejected brine mixes over a relatively deep layer, whereas the melt accumulates within a thin surface layer), which further stabilizes the near-surface ice-ocean system (e.g. Goosse and Zunz 2014). Here, there is sufficient subsurface warmth that sea ice formation remains modest (e.g. Martinson 1990) and that haline coastal or open-ocean deep water formation does not replace the thermal open-ocean convection simulated at 800 ppm.

We have added a new figure of temperature-salinity properties in the sinking regions. This helps to clarify the role of freshwater forcing in the hot case (1600 ppm), and a change in regime in the cold case (400 ppm). We agree that when the surface approaches freezing, the seasonal density forcing undergoes a shift due to a reduced thermal expansion coefficient. We have included further discussion of this with reference to the above papers.

**Technical corrections:**

p1, L17: "We employ an ocean resolution of... " -> "The atmosphere and ocean horizontal resolutions are... respectively."

This has been changed, though we prefer to mention the ocean first.

p1, L19; p8, L4: "simulate the model" -> "run the model".

This has been updated.

p1, L20: "CO2" -> "atmospheric CO2".

"atmospheric" has been added

p1, L26: "salinities" -> "surface salinities".

"surface" has been added to both references to salinity in this sentence.

p2, L10: "role of the ocean circulation" -> "the role of ocean circulation".

This has been updated.

p2, L11: "leading theories of what caused" -> "prevailing proposed mechanisms for". Should albedo feedbacks be mentioned as essential to these mechanisms?

The suggested edit has been made, and a sentence added on the importance of ice albedo feedbacks.

p2, L16: "atmospheric CO2" -> "atmospheric CO2 (thereafter referred to as CO2)".

This has been updated.

p3, L18: Delete "data".

"data" has been deleted.

p5, L3: "it does not capture the subsequent" -> "early and late Eocene topographies differ significantly due to".

This has been done.

p5, L19: "a necessity of" -> "dictated by".

This has been updated.

p6, L12: "future perturbations" is somewhat unclear/ambiguous. I would delete the unnecessary second half of the sentence ("and any. . .. late Eocene").

We agree – since we did not specify what "future perturbations" we had in mind, we have removed that statement.

p6, L15: "the vertical diffusivity" -> "parameterized oceanic vertical mixing".

This has been updated.

p6, L16-17: "mixing is set to a constant background value and then enhanced in the

vicinity of rough-topography" -> "the diffusivity is set to a constant background value and enhanced near the bottom".

This has been updated.

p8, L6-8: "since the BL scheme is. . ." -> "since the former is more physical".

We have changed this to "since the former is more physically realistic".

p8, L8-10: Delete this last sentence of the paragraph, which is somewhat misleading.

We have deleted this sentence.

p10, L14: Delete "similarly".

Deleted.

p11, L4: "cannot" -> "does not". Here and elsewhere, it is argued that salinities are too low for convection to occur. More cautious statements recognizing that what matters is the low salinity relative to that of other high-latitude seas (and hence relative to the deep ocean) would be preferable.

We agree and have changed this in the manuscript.

p11, L7: "forces" is perhaps too strong, given that other factors (e.g. Nilsson et al. 2013) also play a role?

We have clarified this statement to say that it has a freshening influence in the North Pacific.

p12, L12-14: Not clear to me, please explain why.

The areal extent of the western Pacific warm pool in our model is larger than the present day. We suggest that this is partly due to fewer land barriers in the western Pacific and a wider basin. We argue that this creates a larger thermal inertia in the western Pacific, implying a reduction of variance in the east to west thermal gradient.

p13, L21-22: Delete "Both of these sectors. . .tracer distribution."

Deleted.

p13, L22: "The shallow gateways. . .provide barriers": rephrase.

We have rewritten this to indicate that the shallow gateways through Drake Passage and Tasman Seaway inhibit the exchange of deep water between the basins.

p14, L1-2: It is not sufficient that the cell is "warmer and saltier". What matters is the temperature contrast traversed by the circulation.

We agree and have adjusted this statement to mention the temperature contrast.

p14, L3: "due to warm, salty water masses": unclear.

This has been rewritten.

p14, L6-7: Delete "The shape of. . . modern geography".

We have clarified this sentence to say that the Bryan-Lewis scheme simply enhances deep ocean vertical mixing relative to upper level mixing.

p14, L8-10: Rephrase.

We have separated the two clauses into separate sentences and made clearer that we are talking about a priori expectations.

p14, L14: "roughness" -> "bottom-enhanced". Roughness is constant in your implementation.

This has been changed.

p14, L 15-16: "a larger separation. . .. theoretical predictions" -> "a weaker diffusive heat penetration into the abyss".

This has been changed.

p15, L16: "apsects" -> "aspects".

This has been fixed.

p16, L5: "for example higher albedo": not obvious, given potentially stronger albedo feedbacks.

We have clarified with model diagnostics that our net incoming shortwave radiation is indeed higher than the modern case.

p19, L7: "clear" -> "immediate".

We have changed this to "large", since we wish to emphasise the magnitude of change rather than the rate.

p20, L17: "The" -> "the".

Done.

p22, L4: "bottom roughness" -> "bottom-enhanced".

Done

p22, L8: "freshwater" -> "fresh water".

Done

p22, L9-10: "even in the absence of tectonic barriers in the Nordic seas": what do you mean here?

This clause has been removed, since it was superfluous and did not flow properly.

Figures 6 and 12: Showing the MOC of the three (or four) runs in the same figure, in absolute values rather than as differences, would be much clearer.

We have altered the MOC figures to show the total circulation rather than the anomalies.

---

## Author Comment (AC2) · 9 May 2018

We thank the reviewer for his/her thoughtful and constructive comments, which have helped to improve the manuscript. We present the reviewer's comments in black text, and our response in blue text.

**Summary:**

First of all, I did not read the other online review to this paper prior to writing my own, so this review is completely independent.

This paper aims to explore climate sensitivity and ocean overturning at the Eocene-

Oligocene, as well as other aspects of the Earth system. It argues that by using more realistic paleogeographies and a relatively high resolution ocean model, it represents an advancement over previous work. In essence, it describes a set of simulation at 3 $CO_2$ levels for the late Eocene, with 2 different parameterisations of ocean mixing, and includes a model-data comparison. Overall, the paper provides an interesting summary of the basic state and climate sensitivity of this particular model, and is a solid baseline from which additional sensitivity studies can be carried out.

We thank the reviewer for the overall positive assessment.

**Major comments:**

The introduction is a nice overview of some previous modelling work. However, more quantitative information could be presented. This paper focusses mostly on climate sensitivity and overturning, so a summary table of e.g. climate sensitivities and ocean overturning states at different $CO_2$ levels in previous work would be very informative and provide more motivation and context for the work presented here (especially if the paleogeogs and ocean resolution were included in the same table).

We have added a table of previous EOT model simulations, include where possible their climate sensitivities, overturning states, and their ocean and atmosphere resolutions.

Section 2 on the model is good, but it appears that several changes have been made (to e.g. the resolution) compared with the published version of the model. As such, it is important to present an assessment of the performance of the model under preindustrial conditions. In addition, at several points in the manuscript comparisons are made with modern, and these are rather meaningless unless they are comparisons with a modern model simulation from the same model.

We acknowledge that it would be better to compare our model with a control climate model at the same resolution. We are currently developing a new pre-industrial simulation using the same resolution, but this will be a very short run and relatively untested
at the time of revising the paper. We will in any case try to make this comparison as best we can, based on a short time series.

The spinup plots are useful and interesting, but it would be very useful to see Gregory plots of the three simulations as well, to assess how spun up the simulations are, and the likely equilibration temperature of the simulations.

We have added Gregory plots of the spinup evolution in order to assess the projected equilibrium temperatures.

**Specific Comments:**

Abstract: Is the model resolution really greater than previous work? A summarising table of previous simulations would help, see above.

We have added a summary table of previous simulations.

P1, Line 25; P22, line 1: what is meant by "robust" in this context?

We mean robust to $CO_2$ perturbations, but since that point is already explained in the second half of the sentence, we have removed the word 'robust'.

P2, Line 14: for a review of the mechanisms see Lear (2016, Science)

This reference has been added, with an extra sentence on the importance of paleo-geographic boundary conditions.

P2, line 21: seasonality as well as climate sensitivity was important in the Gasson paper.

This point has been added.

P2, line 22: make it clear whether $CO_2$ estimates are for just prior to, or just after the EOT.

We have clarified that observations are based on the late Eocene.

P6, line 10: need to make it clear that although using "realistic" paleotopographies is a step forward compared with some previous studies, there are still uncertainties in these that are not captured in this study.

We have added further description of the uncertainties in the Baatsen et al (2016) reconstruction.

P9, your experiences with model spinup are very interesting and I think warrant inclusion in the conclusions section, and maybe even the abstract.

A summary of the spinup has been included in the conclusions section as suggested.

P9, line 15: a graph illustrating the fully coupled versus asynchronous spinup procedure (e.g. a graph of time against global mean temperature at a couple of ocean depths) would be very interesting and illuminating. i.e. add non-asynchronous run to Figure 2.

The synchronous vs asynchronous spinup evolution has been added to Figure 2.

P9, line 23: add another plot which is the evolution of ocean overturning strength in each basin over the model spinup period.

Evolution of the overturning has been added to the spinup plot.

P9, line 23: Also add another timeseries which is ocean salinity over time in the Arctic (and maybe other ocean basins). It would be interesting to know if this is still decreasing or whether it has reached an equilibrium.

Evolution of surface salinity in the Arctic basin has been added. We have also included indicative values of Pacific and Atlantic salinity.

P11, line 27; p21, line 19: "La Nina-like mean state" I would avoid this term; it has caused much confusion in the Pliocene community. Instead, just say that in the annual mean the west-east gradient in increased.

We have removed the "La Niña-like mean state" reference and rephrased as suggested.

P12, line 26: I don't think that the level of confidence in the interpretation of the proxy data justify this statement. I don't think the data really allows us to say which modelled seaice distribution is best (i.e. there is some seaice at 800ppmv).

We have rephrased this statement with reference to other evidence that suggests warm winters in the Eocene (i.e. less prone to sea ice).

P14, line 5: It would be good to note whether these sensitivity studies to mixing scheme were run for exactly the same length of time and had the same spinup process as the 'standard' runs.

They were run for the same time and with the same spinup procedure. This is now added to the text.

P16, line 4 – for the comparison with modern climate sensitivity, is this exactly the same model and same resolution etc?

This did not use the same resolution. We are currently developing a pre-industrial simulation with the same resolution as our Eocene run, but this will be a short run at the time of re-submitting the manuscript.

P16, line 16: "We suggest that the ice-free conditions allow for substantial radiative warming of SST in addition to atmospheric polar amplification." I don't understand what this means.

We have rephrased this statement. We mainly wanted to mention the two separate effects of radiative forcing and enhanced energy transport that combine to give polar amplification (Alexeev et al, 2005). However, we neglected to mention possible cloud feedbacks, and this has now been added.

Section 4.4: This section is somewhat superfluous, and doesn't add much to the

manuscript unless the location of late Eocene palms are plotted on Figure 13.

We agree and have removed this section.

**Technical comments:**

Figure 1a – get rid of the 'hole' at the north pole and 0 degrees east.

We have interpolated the Arctic region onto a regular lat-lon grid to remove the gaps.

Figure 3 caption – state the $CO_2$ concentration.

This has been added.

Coloured circles in figure 8 are difficult to see – put a black line around the circumference.

Black outlines have been added to the coloured dots.

P15, section 4 title – $CO_2$ not CO2.

We have included the subscript in the title.

Figure 9 would be better presented as both plots being a warming, i.e. 1600-800 and 800-400. In fact, throughout the paper I would prefer presenting the simulations as 3 in a series, rather than one in the middle with a warmer and a cooler either side, to aid consistency.

We have adjusted the plots to show a warm-cold comparison in both cases, as suggested.